# A compartmentalized signaling network mediates crossover control in meiosis

Liangyu Zhang[1,2,3,4], Simone Köhler[1,2,3,4], Regina Rillo-Bohn[1,2,3,4], Abby F Dernburg[1,2,3,4]*

[1]Department of Molecular and Cell Biology, University of California, Berkeley, Berkeley, United States; [2]Howard Hughes Medical Institute, Chevy Chase, United States; [3]Biological Systems and Engineering Division, Lawrence Berkeley National Laboratory, Berkeley, United States; [4]California Institute for Quantitative Biosciences, Berkeley, United States

**Abstract** During meiosis, each pair of homologous chromosomes typically undergoes at least one crossover (crossover assurance), but these exchanges are strictly limited in number and widely spaced along chromosomes (crossover interference). The molecular basis for this chromosome-wide regulation remains mysterious. A family of meiotic RING finger proteins has been implicated in crossover regulation across eukaryotes. *Caenorhabditis elegans* expresses four such proteins, of which one (ZHP-3) is known to be required for crossovers. Here we investigate the functions of ZHP-1, ZHP-2, and ZHP-4. We find that all four ZHP proteins, like their homologs in other species, localize to the synaptonemal complex, an unusual, liquid crystalline compartment that assembles between paired homologs. Together they promote accumulation of pro-crossover factors, including ZHP-3 and ZHP-4, at a single recombination intermediate, thereby patterning exchanges along paired chromosomes. These proteins also act at the top of a hierarchical, symmetry-breaking process that enables crossovers to direct accurate chromosome segregation.
DOI: https://doi.org/10.7554/eLife.30789.001

*For correspondence:
afdernburg@lbl.gov

## Introduction

Meiosis gives rise to haploid gametes through two sequential rounds of nuclear division. In most eukaryotes, accurate meiotic chromosome segregation requires that every pair of homologous chromosomes attains at least one crossover (CO) recombination product, which creates a stable interhomolog connection known as a chiasma (*Page and Hawley, 2003*). However, the total number of COs per cell is typically far too low to assure CO formation on each chromosome by a Poisson process; indeed, in many species each homolog pair undergoes only a single CO (*Mercier et al., 2015*). In cases where multiple COs occur per chromosome pair, they are nonrandomly far apart, a phenomenon known as 'crossover interference' (*Muller, 1916*; *Sturtevant, 1915*). Despite our longstanding awareness of crossover patterning, the mechanisms that ensure this highly nonrandom process remain poorly understood and controversial.

Meiotic recombination is initiated by programmed DNA double-strand breaks (DSBs) catalyzed by the conserved topoisomerase-like enzyme Spo11 (SPO-11 in *C. elegans*) (*Dernburg et al., 1998*; *Keeney, 2008*; *Keeney et al., 1997*). A subset of DSBs is processed to become COs, while the rest are repaired through alternate pathways. In several model organisms, two CO pathways have been elucidated: The 'class I' pathway, which is subject to CO assurance and interference, requires two meiosis-specific homologs of the MutS bacterial mismatch repair protein, Msh4 and Msh5, which form a complex known as MutSγ. Components of the synaptonemal complex (SC), an ordered, periodic proteinaceous structure that assembles between homologous chromosomes during meiotic

**eLife digest** Most human cells contain 23 pairs of chromosomes, giving 46 chromosomes in total. When a cell divides, it typically copies all its chromosomes and distributes the copies equally between the two new cells, so that they also have 46 chromosomes. Cells in the reproductive organs undergo a special division process called meiosis, which halves the number of chromosomes. As a result, sperm and eggs have just 23 chromosomes, and so when they combine, the fertilized egg receives a complete set of 46 chromosomes.

A similar process happens in all species that use sexual reproduction. As the chromosomes prepare to separate, they line up side by side in matching pairs. During this period, DNA from one chromosome will swap with DNA from its partner. These events are called "crossovers," and because such exchanges can happen at many locations along the chromosomes, no two sperm or eggs are the same.

Historic studies in fruit flies revealed that chromosomes do not mix their DNA at random. After one crossover occurs, it is less likely that another will happen, and if there are two crossovers, the second one tends be far away from the first. This suggests that there must be a signal that tells the chromosomes about the exchange. However, the nature of the signals and how they are communicated along pairs of chromosomes remain mysterious.

A structure called the "synaptonemal complex" holds chromosomes together while they mix their DNA. In 2017, researchers found that this structure behaves like a liquid crystal: its molecules are organized into a regular, repeating pattern, but they move freely, like a fluid. If signals could move through this material, this might explain how information spreads along paired chromosomes.

Now, Zhang et al. – including two researchers involved in the 2017 work – identify some of the signals in the small roundworm, *Caenorhabditis elegans*, as four related proteins named ZHPs. When tagged with fluorescent markers and followed under a microscope, all four ZHP proteins moved to the liquid crystal-like synaptonemal complex during meiosis. Depleting the proteins at this crucial time revealed their roles. Two of the proteins are needed for chromosomes to mix their DNA, while the other two control the number of exchanges between each pair of chromosomes. Successful meiosis depended upon all four ZHPs, and so too did the fertility of the worms.

The next step is to find the other molecules that interact with the ZHP proteins during meiosis. Since similar proteins appear in other species, including humans, this could help to reveal more about how genetic traits from our parents mix and match. In the future, studies that build on these findings could also help scientists to understand how errors in these processes give rise to birth defects and infertility.

DOI: https://doi.org/10.7554/eLife.30789.002

prophase, are also required for class I COs. An alternate enzymatic pathway can give rise to both noncrossovers and 'class II' COs, which do not show interference.

In the nematode *C. elegans*, only class I COs normally occur, and only a single CO occurs per chromosome pair (*Martinez-Perez and Colaiácovo, 2009*). No COs occur in the absence of SC assembly (*MacQueen et al., 2002*). Additional factors required for meiotic COs, but not for homolog pairing, synapsis, or other homologous recombination, include MSH-4 and MSH-5 (*Kelly et al., 2000*; *Zalevsky et al., 1999*), the cyclin-related protein COSA-1 (*Yokoo et al., 2012*), and ZHP-3, a RING finger protein (*Jantsch et al., 2004*).

COs act together with sister chromatid cohesion to orchestrate two successive rounds of chromosome segregation. In mammals and most other model organisms, cohesion between sister chromatids is maintained near centromeres during the first division (MI), but arm cohesion must be released to allow homologs to separate (*Duro and Marston, 2015*). In *C. elegans*, which lacks defined centromeres, cohesion is spatially regulated downstream of CO designation. Late in meiotic prophase, the chromosome region on one side of the designated CO site becomes enriched for several proteins, including HTP-1/2, LAB-1, and REC-8, which act together to maintain cohesion through the first division. Conversely, the reciprocal chromosome region becomes depleted of these proteins but retains a stretch of SC. This 'short arm' eventually recruits the Aurora B kinase AIR-2, which is required to release cohesion and allow MI segregation (*de Carvalho et al., 2008*; *Ferrandiz et al., 2018*;

*Kaitna et al., 2002*; *Martinez-Perez et al., 2008*; *Nabeshima et al., 2005*; *Rogers et al., 2002*; *Severson and Meyer, 2014*). It has been unclear whether the SC that remains along this arm contributes to regulating cohesion. The hierarchy of steps leading to this remodeling, and particularly the initial trigger for this asymmetry, are also poorly understood.

Several recent studies have indicated that structural proteins within assembled SCs are highly mobile (*Nadarajan et al., 2017*; *Pattabiraman et al., 2017*; *Rog et al., 2017*; *Voelkel-Meiman et al., 2012*). We have also presented evidence that this material likely assembles through regulated phase separation (*Rog et al., 2017*). When SCs cannot assemble between chromosomes, as in *C. elegans* mutants lacking the essential chromosome axis protein HTP-3, or in budding yeast lacking DSBs, SC proteins self-assemble to form irregularly shaped, but internally structured, nuclear bodies known as polycomplexes (*Roth, 1966*), which do not appear to be associated with chromosomes. Observations of *C. elegans* polycomplexes *in vivo* revealed that they behave as liquid-like droplets, in that they rapidly change shape and fuse to form larger bodies (*Rog et al., 2017*). Taken together, these observations indicate that SC proteins self-assemble through regulated coacervation to form a liquid crystalline material – defined as an ordered assembly of molecules that diffuse freely relative to each other. Normally this material preferentially assembles as a bilaterally symmetrical lamina between paired axes, but in the absence of the proper chromosomal substrate, it forms polycomplexes.

Growing evidence has implicated the SC in CO regulation, particularly in *C. elegans* (*Hayashi et al., 2010*; *Libuda et al., 2013*; *MacQueen et al., 2002*; *Rog et al., 2017*; *Sym and Roeder, 1994*). The mobility of proteins within assembled SCs suggests that biochemical signals such as enzymes or posttranslational modifications could move through this material to regulate CO formation along paired chromosomes. In support of this idea, a number of crossover factors and DNA damage repair components have been found to associate with both SCs and polycomplexes. In budding yeast, these include Msh4, Msh5, Spo16, Spo22 (Zip4), Zip2, and Zip3, and the 9-1-1 complex component Mec3 (*Shinohara et al., 2015*; *Tsubouchi et al., 2006*). Zip3 is unique among these proteins in that it localizes throughout the body of polycomplexes, suggesting that it has an intrinsic affinity for SC proteins, while all other proteins reported to date appear to localize to a 'cap' or focus at the surface of polycomplexes. We observed that ZHP-3, a *C. elegans* homolog of yeast Zip3 (*Agarwal and Roeder, 2000*), similarly localizes throughout polycomplexes (*Rog et al., 2017*). Interestingly, ZHP-3 and Zip3 share homology with Hei10 and RNF212, which have been implicated in crossover control across diverse eukaryotes (*Chelysheva et al., 2012*; *De Muyt et al., 2014*; *Gray and Cohen, 2016*; *Qiao et al., 2014*; *Reynolds et al., 2013*; *Wang et al., 2012*; *Ward et al., 2007*). All of these proteins contain an N-terminal RING finger domain, a central coiled-coil domain, and a C-terminal domain that is predicted to be largely unstructured.

Three paralogs of ZHP-3 are expressed in the *C. elegans* germline, but their functions have not previously been described. Here we define essential meiotic roles for these proteins. We report that this family of four paralogs, which we designate as ZHP-1–4, likely act as two heterodimeric complexes to regulate CO formation. Like ZHP-3, the other 3 proteins localize throughout SCs in early prophase, and are also recruited to polycomplexes. Together with other known and unknown factors, these ZHP proteins appear to form a signaling network that acts within the SC to ensure CO formation while limiting the number of CO-designated sites, and may thus mediate both CO assurance and CO interference. In addition, the ZHP proteins in *C. elegans* act upstream of other regulators to direct chromosome remodeling in response to crossover formation, which enables the stepwise removal of cohesion that enables two successive rounds of chromosome segregation during meiosis. Homology between the ZHP proteins and meiotic regulators from other phyla indicate that similar mechanisms likely underlie meiotic CO control in most eukaryotes, and may also play previously-unrecognized roles in regulating sister chromatid cohesion during meiosis.

## Results

### A family of meiotic RING finger proteins in *C. elegans*

As described in the Introduction, the RING finger protein ZHP-3 was initially identified as a candidate meiotic factor in *C. elegans* based on its homology to Zip3, a component of the 'synapsis initiation complex' in yeast (*Agarwal and Roeder, 2000*). Targeted disruption of *zhp-3* revealed that it is

required for CO formation, but dispensable for homolog pairing and synapsis (*Jantsch et al., 2004*). The *C. elegans* genome includes three additional predicted genes with the same domain structure and significant homology to ZHP-3 (*Figure 1—figure supplement 1A–B*). Several independent analyses indicate that all four genes are expressed in the germline, consistent with a role in meiosis (WormBase; (*WormBase, 2017*); S.K. and A.F.D. unpublished).

All four predicted proteins have similar N-terminal C3HC4-type RING-finger domains (*Figure 1—figure supplement 1A–B*), which are usually associated with ubiquitin E3 ligase activity (*Deshaies and Joazeiro, 2009*). The C-terminal regions of these proteins are divergent and lack obvious structural domains, but contain a number of potential post-translational modification sites (*Figure 1—figure supplement 1A*). Based on their similarity to each other, and on evidence presented here that they function together to form a regulatory circuit, we have named these genes *zhp-1* (F55A12.10), *zhp-2* (D1081.9) and *zhp-4* (Y39B6A.16) (*zip3* homologous protein). This numbering reflects their physical order in the *C. elegans* genome: *zhp-1,-2*, and *-3* are clustered on Chromosome I, while *zhp-4* is on Chromosome V.

To gain insight into the evolution of these proteins, we searched for homologs in other sequenced nematode genomes (*Figure 1—figure supplement 1C*). Orthologs of each protein can be identified in *C. briggsae* and *C. remanei*, as well as in other nematode genera, indicating that these proteins diversified at least tens to hundreds of millions of years ago (*Figure 1—figure supplement 1C–D* and data not shown). Their N-terminal regions show homology to the mammalian CO regulator RNF212, and ZHP-1 and ZHP-2 are also recognized by BLAST as homologs of HEI10, which is involved in CO regulation in mammals, plants, and fungi (*Chelysheva et al., 2012*; *De Muyt et al., 2014*; *Qiao et al., 2014*; *Rao et al., 2017*; *Reynolds et al., 2013*, *Wang et al., 2012*; *Ward et al., 2007*) (data not shown).

## ZHP proteins exhibit two distinct patterns of dynamic localization during meiosis

To investigate the ZHP proteins, we inserted epitope tags at the C-terminus of each coding sequence; these tagged proteins supported normal meiosis (*Supplementary file 1*, Table S1) and were readily localized by immunofluorescence with epitope-specific antibodies (*Figure 1—figure supplement 2* and *Figure 2—figure supplement 2A–B*). At meiotic entry the ZHP proteins localized to small polycomplexes, which appear prior to synapsis and contain the four known SC structural proteins, SYP-1,-2, -3, and -4 (*Goldstein, 2013*; *Rog et al., 2017*; *Roth, 1966*). Once SC assembly initiated, they localized to SCs (*Figure 1*).

As meiosis progressed, the four proteins showed two distinct patterns of localization (*Figure 1*): From early prophase to mid-pachynema, ZHP-1 and -2 became brighter and more contiguous along the length of SCs, although the intensity of staining was not homogeneous. Upon the appearance of GFP-COSA-1 foci, which mark designated CO sites from mid-pachynema through diplonema (*Yokoo et al., 2012*), ZHP-1 and -2 became confined to the SC on one side of each CO. Intriguingly, this restriction was observed in pachytene nuclei that retained SC (*Figure 1A–B*) and the HORMA domain proteins HTP-1/2 (data not shown) on both sides of the CO, and represents the earliest known molecular differentiation of the two chromosome domains that will become the long and short arms of the bivalent. ZHP-1 and -2 remained associated with SC proteins along the short arm of each bivalent as the SC disappeared from the long arm, and persisted as long as SC proteins were present along the short arms, through late diakinesis.

ZHP-3 and -4 exhibited a similar distribution to ZHP-1 and -2 during early prophase, but showed a distinct pattern starting at mid-pachytene, when crossover designation (the selection of one recombination intermediate as the eventual crossover site for each pair of chromosomes) is thought to occur in *C. elegans* (*Figure 1C–D*) (*Yokoo et al., 2012*). Upon the detection of GFP-COSA-1 foci, which mark the subset of recombination intermediates that will become COs, ZHP-3 and -4 both accumulated at these sites and gradually disappeared from the SC along both arms of the chromosomes. These observations are largely consistent with previous characterization of ZHP-3 and a partially functional GFP-ZHP-3 fusion protein (*Bhalla et al., 2008*; *Jantsch et al., 2004*).

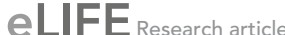

**Figure 1.** ZHP proteins exhibit two distinct patterns of dynamic localization. (A–D) Projection images showing immunofluorescent localization of 3xFLAG tagged ZHP-1 (A), ZHP-2 (B), ZHP-3 (C) and ZHP-4 (D) relative to SCs (marked by SYP-1) and CO sites (marked by GFP-COSA-1) from mitosis to early diakinesis in the germ line. Insets in each panel show representative images of ZHP localization in (left to right) early pachytene (EP), mid-pachytene (MP), late pachytene (LP) and diplotene (Di) nuclei. Inserts show diagrams to clarify the localization patterns of ZHPs (red) from MP to Di.

*Figure 1 continued on next page*

**Figure 1 continued**

Gray lines, blue line and green focus represent the chromosome axis, central region of the synaptonemal complex (SC) and CO site, respectively. ZHP-1 and ZHP-2 show identical dynamics: they localize to assembling SCs, and upon the appearance of COSA-1 foci they become restricted to the short arm of each bivalent. ZHP-3 and ZHP-4 also mirror each other, localizing initially throughout the SC, and then accumulating at one site along each chromosome pair, concomitant with the appearance of COSA-1 at these sites. By late pachynema, 100% of GFP-COSA-1 foci colocalize with ZHP-4 puncta (n = 2,700 COSA-1 foci in 450 nuclei from eight gonads [11 rows of cells per gonad] were scored). 3F indicates 3xFLAG. Scale bars, 5 μm.

DOI: https://doi.org/10.7554/eLife.30789.003

The following figure supplements are available for figure 1:

**Figure supplement 1.** Sequence alignment of *C. elegans* ZHPs, and the evolution of ZHPs in nematodes.

DOI: https://doi.org/10.7554/eLife.30789.004

**Figure supplement 2.** Absence of nonspecific staining with anti-FLAG antibodies in the *C. elegans* germline.

DOI: https://doi.org/10.7554/eLife.30789.005

**Figure supplement 3.** Localization of the ZHPs along meiotic chromosomes depends on SCs.

DOI: https://doi.org/10.7554/eLife.30789.006

## Localization of ZHPs along meiotic chromosomes depends on synaptonemal complexes

The SC in *C. elegans* comprises at least four proteins, known as SYP-1–4, which are mutually dependent for assembly between chromosomes or as polycomplexes. We found that none of the four ZHP proteins associated along chromosomes in *syp-1* or other *syp* mutants (*Figure 1—figure supplement 3A–C*, and data not shown). Intriguingly, in the region of the germline that normally corresponds to late pachynema, the four ZHP proteins colocalized within nuclear puncta devoid of SC proteins, indicating that the ZHPs can interact with each other in the absence of SCs or polycomplexes (*Figure 1—figure supplement 3C*, lower panel). These puncta do not coincide with known DSB repair proteins (*Figure 1—figure supplement 3D*). The ZHP proteins were also less abundant in *syp-1* mutants (*Figure 1—figure supplement 3E–K*), suggesting that their stability is enhanced by association with SCs, particularly that of ZHP-1 and -2 (*Figure 1—figure supplement 3F and J*).

Because the ZHP proteins relocalized upon CO designation (*Figure 1*), we examined their distribution under conditions where COs fail to occur. In mutants lacking SPO-11, MSH-5, or COSA-1, ZHPs were detected along the length of SCs through diplonema (data not shown). Thus, relocalization of the ZHPs in late prophase depends on DSBs and pro-CO factors.

## ZHPs likely act as two heterodimeric complexes

To investigate the meiotic functions of the ZHP proteins, we first exploited the auxin-inducible degradation system. Each of the ZHP genes was tagged with a 3xFLAG epitope or GFP and a 44-aa degron sequence (hereafter 'AID') in a strain that expresses the F-box protein AtTIR1 throughout the germline (*Zhang et al., 2015*). When adult animals were transferred to plates containing 1 mM auxin, all AID-tagged proteins were robustly depleted within 1–2 hr (*Figure 2—figure supplement 1A–B*). To assess the efficacy of the knockdown, we compared meiotic chromosome segregation in hermaphrodites carrying the previously characterized *zhp-3(jf61)* null allele (*Jantsch et al., 2004*) to auxin-treated worms expressing ZHP-3-AID-GFP (*Figure 2—figure supplement 1C*). AID-mediated depletion of ZHP-3 quantitatively phenocopied the null allele with respect to both embryonic viability (Δ*zhp-3*: 2.06 ± 0.62%; *zhp-3::AID*: 1.83 ± 0.82%) and the incidence of males among the self-progeny of hermaphrodites, which reflects X-chromosome nondisjunction (*Hodgkin et al., 1979*) (Δ*zhp-3*: 43.75 ± 27.59%; *zhp-3::AID*: 37.02 ± 38.12%) (*Figure 2—figure supplement 1C*), indicating that auxin-induced degradation effectively eliminates the function of ZHP-3.

To determine the dependence of the ZHP proteins on each other for their localization and stability, we exposed worms to auxin for 24 hr, so that a pool of nuclei had entered and progressed through meiotic prophase in the absence of AID-tagged protein. We found that ZHP-1 and -2 depend on each other for both their chromosome association and stability (*Figure 2A–D*). ZHP-3 and -4 were also interdependent for their localization to chromosomes (*Figure 2E and G* and *Figure 2—figure supplement 2C*). ZHP-3 was unstable in the absence of ZHP-4 (*Figure 2F*) and ZHP-4 levels were somewhat lower when ZHP-3 was depleted (*Figure 2H* and *Figure 2—figure supplement 2D*).

**Figure 2.** ZHPs likely act as two heterodimeric protein complexes. (**A–H**) Projection images of mid-pachytene and diplotene nuclei, showing the localization of ZHPs (green), with corresponding Western blots to assess protein levels in the presence and absence of their partners. SCs are marked by SYP-1 (red). Tubulin was blotted as a loading control. Localization of each ZHP to SCs was abolished when its partner was depleted by auxin treatment for 24 hr. In the absence of their partners, the ZHPs were also destabilized. Scale bars, 5 μm. (**I**) Yeast 2-hybrid interactions monitored by β-

*Figure 2 continued on next page*

*Figure 2 continued*

galactosidase expression reveal specific interactions between ZHP-1 and ZHP-2, and ZHP-3 and ZHP-4, respectively. No interactions were detected for other combinations. 'Empty' indicates the same vector with no insert. None control: pPC97 and pPC86 with no insert; weak control: pPC97-human RB amino acids 302–928 and pPC86-human E2F1 amino acids 342–437; moderate control: pPC97-*Drosophila* DP amino acids 1–377 and pPC86-*Drosophila* E2F amino acids 225–433; and strong control: pPC97-rat cFos amino acids 132–211 and pPC86-mouse cJun amino acids 250–325.

DOI: https://doi.org/10.7554/eLife.30789.007

The following figure supplements are available for figure 2:

**Figure supplement 1.** Depletion of ZHP proteins using the AID system.

DOI: https://doi.org/10.7554/eLife.30789.008

**Figure supplement 2.** Additional antibody validation, ZHP-4 in *zhp-3* mutants, and yeast 2-hybrid analysis of ZHP protein interactions.

DOI: https://doi.org/10.7554/eLife.30789.009

Other RING finger proteins such as BRCA1/BARD1 are known to act as heterodimers (*Brzovic et al., 2001*; *Metzger et al., 2014*; *Wu et al., 1996*). The similarity and interdependence of localization of ZHP-1 and -2, and of ZHP-3 and -4, suggested that these four proteins might function as two pairs. Using yeast two-hybrid analysis, we found that ZHP-1 and -2 indeed interact with each other, and that ZHP-3 and -4 also physically interact (*Figure 2I* and *Figure 2—figure supplement 2E*). This evidence for specific pairwise physical interactions, together with their physical distributions (above) and functional analysis (below), indicate that ZHP-1 and -2 form a complex, and that ZHP-3 and -4 similarly act as a pair. Hereafter we refer to these two dimeric activities as ZHP-1/2 and ZHP-3/4 for simplicity.

## ZHPs play essential roles in chiasma formation and chromosome segregation

Using a number of established assays, we investigated the meiotic functions of these RING finger proteins. At late diakinesis, association between homologs is normally maintained by chiasmata, and six condensed bivalents can be detected in each wild-type oocyte nucleus. We exposed *zhp-AID* strains to auxin for 24 hr and then analyzed chiasma formation. Depletion of ZHP-1 or ZHP-2 yielded 8–12 DAPI-staining bodies at diakinesis, representing 4–12 univalents and 0–4 bivalents (*Figure 3A–B*). In contrast, depletion of ZHP-3 or ZHP-4 resulted in a complete failure of chiasma formation, indicated by the appearance of 12 achiasmate chromosomes in each nucleus (*Figure 3C–D*), consistent with previous characterization of ZHP-3 (*Bhalla et al., 2008*; *Jantsch et al., 2004*).

Because a few chiasmata were observed when ZHP-1-AID or ZHP-2-AID were depleted, we were concerned that these proteins might not be efficiently degraded. We therefore introduced null mutations in *zhp-1* and *zhp-2* (see Materials and methods). Homozygous mutants produced few viable progeny and a high incidence of males among the self-progeny of hermaphrodites (*Figure 3E*), both of which are indicative of meiotic chromosome missegregation. They also showed identical distributions of DAPI-staining bodies at diakinesis to that observed in AID-depleted animals (*Figure 3—figure supplement 1A–C* and data not shown). Additionally, both *zhp-1* and *-2* null mutants produced more viable progeny than *zhp-3* null mutants (***p<0.001, two-sided Student's *t*-test) (*Figure 3E*), consistent with a less absolute dependence of chiasma formation on ZHP-1/2 than on ZHP-3/4.

We wondered whether the few chiasmata in hermaphrodites lacking ZHP-1/2 might preferentially occur on the X chromosome, which shows some differences from autosomes in its genetic requirements for DSB induction and CO formation (*Yu et al., 2016*). To investigate this, we used fluorescence *in situ* hybridization (FISH) to mark Chromosome V and the X chromosome in *zhp-1* mutants. By examining nuclei at diakinesis, we found that chiasma formation was impaired to a similar degree on both chromosomes. Our observations are quantitatively consistent with a low number of chiasmata distributed evenly among all six chromosome pairs (*Figure 3F–G*).

ZHP-3 was previously found to be dispensable for homologous chromosome pairing, synapsis, and DSB induction (*Jantsch et al., 2004*). Pairing and synapsis also occurred normally in the absence of ZHP-1, -2, or -4 (*Figure 3—figure supplement 2*). RAD-51 foci, which mark recombination intermediates and require DSB induction (*Colaiácovo et al., 2003*), were also abundant in all *zhp* mutants (*Figure 3—figure supplement 3A–F*). Thus, the high frequency of achiasmate





**Figure 3.** ZHP proteins play a central role in chiasma formation and meiotic chromosome segregation. (A–D) Upper: DAPI-stained oocyte nuclei at late diakinesis. Each panel shows a representative nucleus. Lower: Graphs indicating the distribution of DAPI-staining bodies observed at diakinesis. ZHPs tagged with AID in a $P_{sun-1}::TIR1$ background were treated with or without auxin for 24 hr. In the absence of auxin, each nucleus contains six bivalents (homolog pairs held together by chiasmata). Depletion of ZHP-1 or -2 leads to a marked decrease in bivalents per nucleus (8–12 DAPI-staining bodies, or 0–4 bivalents), while depletion of ZHP-3 or -4 results in a complete loss of bivalents (12 DAPI-staining bodies). $n$ is the number of nuclei scored for each condition or genotype. Depletion of ZHP-1,–2, −3 or −4 significantly increase the number of DAPI-staining bodies (***p<0.0001 by Chi-square test for trend). The difference between ZHP-1 depletion and ZHP-2 depletion or ZHP-3 depletion and ZHP-4 depletion is not significant (p=0.7537 and 0.9760, respectively, by Chi-square test for trend). The difference between depletion of ZHP-1 or ZHP-2 and depletion of ZHP-3 or ZHP-4 is significant (***p<0.0001 by Chi-square test for trend). (E) Frequencies of males and viable embryos observed among the whole broods in wild-type, *zhp-1*, *zhp-2* and *zhp-3* null mutant hermaphrodites. (F–G) Sex chromosomes and autosomes show a similar reduction in chiasma formation in the absence of ZHP-1. (F) High magnification images of nuclei at late diakinesis from *zhp-1* heterozygous controls and null mutant hermaphrodites, hybridized with FISH probes recognizing either the right end of X-chromosome or 5S rDNA on Chromosome V, and stained with DAPI. (G) Graph showing the frequency of bivalents observed for Chromosome V and the X Chromosome. 'Expected value' is the frequency for any single chromosome, given the average number of bivalents we observed, and assuming that they were distributed equally among six chromosome pairs. Data were derived from the number

*Figure 3 continued on next page*

*Figure 3 continued*

of DAPI-staining bodies in *zhp-1* heterozygotes and null mutants. Data are represented as mean ±SEM from three independent experiments (*n* = 103 and 145 nuclei, respectively). Significance was tested using the two-sided Student's *t*-test. Scale bars, 5 μm.

DOI: https://doi.org/10.7554/eLife.30789.010

The following figure supplements are available for figure 3:

**Figure supplement 1.** Chiasma formation in *zhp-1* or *zhp-3* null mutants.

DOI: https://doi.org/10.7554/eLife.30789.011

**Figure supplement 2.** ZHP are dispensable for homolog pairing and synapsis.

DOI: https://doi.org/10.7554/eLife.30789.012

**Figure supplement 3.** ZHPs are dispensable for DSB induction and the crossover assurance checkpoint.

DOI: https://doi.org/10.7554/eLife.30789.013

chromosomes in the absence of ZHPs reflects defects in CO recombination downstream of pairing, synapsis, and DSBs.

Prior work has established the existence of a 'crossover assurance checkpoint' in *C. elegans* that delays meiotic progression when recombination intermediates or pro-CO factors are absent on one or more chromosomes due to defects in homolog pairing, synapsis, DSB induction, or recombination (*Carlton et al., 2006*; *Kim et al., 2015*; *Yu et al., 2016*). Activation of this checkpoint prolongs the activity of the CHK-2 kinase, resulting in an extended region and elevated number of RAD-51 foci, and a shift in the pattern of meiotic recombination, but does not perturb CO interference. We observed that phosphorylation of HIM-8 and its paralogs, a marker for CHK-2 activity, was extended when any of the ZHPs was depleted (*Figure 3—figure supplement 3G–I*), and RAD-51 foci were more abundant (*Figure 3—figure supplement 3A–F*). Thus, the crossover assurance checkpoint does not require ZHP activity.

We noted differences in the dynamics of DNA repair when ZHP-1/2 were depleted as compared to ZHP-3/4 (*Figure 3—figure supplement 3A–F*). In the absence of ZHP-3/4, RAD-51 foci reached higher peak numbers than in wild-type animals or in the absence of ZHP-1/2, but depletion of ZHP-1/2 caused a greater delay in both the accumulation and disappearance of RAD-51 foci than depletion of ZHP-3/4. Thus, the two ZHP complexes likely play distinct roles in the processing of recombination intermediates, similar to observations of HEI10 and RNF212 in mouse spermatocytes (*Qiao et al., 2014*).

## ZHP-3/4 are essential to stabilize CO intermediates, while ZHP-1/2 restrict pro-CO activities and promote CO maturation

To probe the role of ZHPs in recombination, we examined the cytological distributions of MSH-5, which can be detected at sites of early recombination intermediates, and COSA-1, which forms bright foci at sites designated for CO formation (*Yokoo et al., 2012*). In wild-type and control (non-auxin-treated) worms, ~10–20 MSH-5 foci per nucleus could be detected by mid-pachytene (*Figure 4—figure supplement 1A*), after which six GFP-COSA-1 foci – one per pair of chromosomes – were observed (*Figure 4A*), and MSH-5 became restricted to these sites (*Figure 4—figure supplement 1*), as previously described (*Yokoo et al., 2012*). In worms depleted of ZHP-3/4, neither MSH-5 nor COSA-1 foci were detected during meiotic prophase (*Figure 4B*, *Figure 4—figure supplement 1B* and data not shown). This indicates that ZHP-3/4 are likely required for stable association of MutSγ with early recombination intermediates. In striking contrast, depletion of ZHP-1/2 resulted in appearance of up to ~50 MSH-5 foci and up to ~30 dim but detectable GFP-COSA-1 foci per oocyte nucleus (*Figure 4C–D*, and *Figure 4—figure supplement 1C–D*). Notably, the number of detectable foci increased in number as nuclei advanced from mid- to late pachytene, and then rapidly declined (*Figure 4D* and *Figure 4—figure supplement 1D*).

The dim GFP-COSA-1 foci in ZHP-1/2-depleted worms showed extensive colocalization with MSH-5 foci (*Figure 4—figure supplement 1E*). Together with prior evidence, this suggested that MSH-5 and COSA-1 colocalize at many recombination intermediates in the absence of ZHP-1/2. Indeed, it is likely that these pro-CO factors initially colocalize at all strand exchange intermediates, and that the difference in the number of detected foci of MSH-5 and COSA-1 is due to the relative sensitivity of the available detection reagents.



**Figure 4.** ZHP-3/4 are required to stabilize recombination intermediates, while ZHP-1/2 promote accumulation of pro-CO factors at a subset of intermediates during late prophase. (A–F) Low-magnification images of mid- and late pachytene nuclei stained for GFP-COSA-1 (green) and SYP-1 (red). Insets on the right showing corresponding nuclei at late pachynema. (A) Representative *zhp-4::AID; P_sun-1::TIR1* germline showing six sites per nucleus marked by bright GFP-COSA-1 foci in the absence of auxin treatment. (B) Depletion of ZHP-4 by auxin treatment for 24 hr resulted in an

*Figure 4 continued on next page*

*Figure 4 continued*

absence of COSA-1 foci. (**C**) Control *zhp-1::AID; P~sun-1~::TIR1* worm showing six bright GFP-COSA-1 foci per nucleus. (**D**) Depletion of ZHP-1 by auxin treatment for 24 hr results in a large number of recombination intermediates marked by dim GFP-COSA-1 foci. Image acquisition and display were adjusted to allow the dim foci to be visualized. (**E, F**) Co-depletion of ZHP-1 and SPO-11 (**E**) or of ZHP-1 and ZHP-4 (**F**) results in an absence of COSA-1 foci. (**G**) Left: Oocyte nuclei at late diakinesis, fixed and stained with DAPI. Each panel indicates a single representative nucleus. Right: Graphs of the distribution of DAPI-staining bodies at diakinesis. Co-depletion of ZHP-1 and ZHP-4 eliminates bivalents (12 DAPI-staining bodies). The number of nuclei scored for each condition was: $n$ = 84 and 104, respectively. (**H**) Quantification of COs in wild type and *zhp-1* null mutant oocytes by whole genome sequencing. $n$ = 8 oocytes for each genotype. **p=0.0072, Mann-Whitney test. Scale bars, 5 µm.
DOI: https://doi.org/10.7554/eLife.30789.014
The following figure supplements are available for figure 4:

**Figure supplement 1.** ZHP-3/4 are required for the appearance of MSH-5 foci, while ZHP-1/2 limit the number of foci during late prophase.
DOI: https://doi.org/10.7554/eLife.30789.015
**Figure supplement 2.** Robust protein co-depletion by the AID system and persistence of ZHP-3 throughout SCs following ZHP-1/2 depletion.
DOI: https://doi.org/10.7554/eLife.30789.016

To validate the idea that these foci correspond to recombination intermediates, we eliminated recombination by disrupting DSB formation. We AID-tagged SPO-11 and either ZHP-1 or ZHP-2 in the same strain, and found that both proteins could be effectively co-depleted by auxin treatment (*Figure 4—figure supplement 2A* and data not shown). This abolished all MSH-5 and GFP-COSA-1 foci (*Figure 4E* and data not shown), supporting the idea that the dim foci observed in the absence of ZHP-1/2 are indeed recombination intermediates. Co-depletion of ZHP-1 and ZHP-4 also eliminated GFP-COSA-1 foci (*Figure 4F* and *Figure 4—figure supplement 2B*) and bivalent formation (*Figure 4G*), supporting the conclusion that ZHP-3/4 are required to stabilize MutSγ at all recombination intermediates. Depletion of ZHP-1/2 also resulted in persistence of ZHP-3 and -4 throughout SCs in late pachytene nuclei (*Figure 4—figure supplement 2C* and data not shown). These observations indicate that ZHP-1/2 are required to restrict pro-CO proteins to a single recombination intermediate along each pair of homologs during late prophase.

We next sought to understand why oocytes lacking ZHP-1/2 have abundant, dim COSA-1 foci but few bivalent chromosomes. This discrepancy could indicate that the dim foci do not mature as COs, or that they do become COs but fail to give rise to stable chiasmata. To address this, we mapped COs in *zhp-1* null mutants by whole genome sequencing (see Materials and methods). Using this genetic recombination assay, an average of 2.8 ± 1.8 (SD) COs per oocyte were detected in *zhp-1* mutants, compared to 7.3 ± 3.0 (SD) per oocyte (slightly more than the expected number of 6) in wild-type animals (*Figure 4H*). These data are consistent with the observed number of bivalents (*Figure 3* and *Figure 3—figure supplement 1*), and indicate that only a small fraction of COSA-1-positive foci in *zhp-1/2* mutants mature to become COs and chiasmata.

The excess number but weak immunofluorescence of GFP-COSA-1 foci in the absence of ZHP-1/2 suggested that one or more limiting components might be distributed to an excess of recombination intermediates. We therefore wondered whether accumulation of pro-CO factors at specific recombination intermediates could be rescued by restricting the number of these intermediates. Mutations in *dsb-2* cause an age-dependent reduction in meiotic DSBs: very few RAD-51 foci and chiasmata are seen in older (48 hr post-L4) *dsb-2* hermaphrodites, while somewhat more RAD-51 foci and chiasmata are seen in younger adults (*Rosu et al., 2013*). We co-depleted DSB-2 and ZHP-2 using the AID system (*Figure 5—figure supplement 1A*), fixed worms at 24 and 48 hr post-L4, and stained for RAD-51 and GFP-COSA-1. Fewer COSA-1 foci and bivalents were observed in animals depleted for both DSB-2 and ZHP-2 than DSB-2 alone, although the number of RAD-51 foci was similar (*Figure 5A–C*, *Figure 5—figure supplement 1B* and data not shown). Intriguingly, while younger *dsb-2* mutant animals have more meiotic DSBs (*Rosu et al., 2013* and data not shown), we observed fewer bright GFP-COSA-1 foci in young ZHP-2 depleted animals: only 11% of nuclei contained a bright GFP-COSA-1 focus, compared to 49% in older animals (*Figure 5B–C*). The number of bivalents in these animals corresponded well with bright GFP-COSA-1 foci, and only chromosomes with bright COSA-1 foci displayed the normal hallmarks of chromosome remodeling (see below), indicating that only bright COSA-1 foci reliably mature as COs.

As another approach to examine how the number of intermediates impacts CO maturation in the absence of ZHP-1/2, we co-depleted ZHP-2 and SPO-11 to abolish programmed meiotic DSBs, and

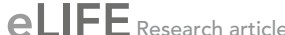

**Figure 5.** CO designation in the absence of ZHP-1/2. (**A**) Projection images of representative late pachytene nuclei stained for GFP-COSA-1 (green) and SYP-1 (red). In the absence of DSB-2 at 24 hr post-L4, an average of 3 bright GFP-COSA-1 foci are detected in each nucleus. However, very few bright GFP-COSA-1 foci are observed when ZHP-2 is co-depleted. By 48 hr post-L4, only ~2 bright GFP-COSA-1 foci are observed per nucleus in the absence of DSB-2, and most nuclei display 0–1 bright GFP-COSA-1 focus when ZHP-2 is also depleted. (**B**) Quantification of bright GFP-COSA-1 foci at late

*Figure 5 continued on next page*

*Figure 5 continued*

pachynema, 24 hr post-L4. Worms of the indicated genotypes were maintained in the presence or in the absence of 1 auxin for 24 hr and then dissected for analysis. Data were derived from 4 to 8 gonads for each genotype or condition. n = 165, 348, 172 and 377 nuclei, respectively. The number of GFP-COSA-1 foci differed significantly between *dsb-2::AID* and *zhp-2::AID; dsb-2::AID* following auxin treatment (***p<0.0001 by Chi-square test for trend). (C) Quantification of bright GFP-COSA-1 foci in late pachytene nuclei 48 hr post-L4. Data were derived from 4 to 8 gonads for each genotype or condition. n = 252, 558, 263 and 416 nuclei, respectively. Depletion of ZHP-2 significantly reduces the number of GFP-COSA-1 foci in the absence of DSB-2 (***p<0.0001 by Chi-square test for trend). The distribution of GFP-COSA-1 foci is also significantly different between 24 hr and 48 hr post-L4 for auxin treated *zhp-2::AID; dsb-2::AID* hermaphrodites (***p<0.0001 by Chi-square test for trend). (D) Representative images of mid-pachytene (MP) and late pachytene (LP) nuclei stained for RAD-51 (yellow), GFP-COSA-1 (green), SYP-1 (red), and DNA (blue). GFP-COSA-1 foci in the absence of ZHP-1/2 was visualized following exposure of SPO-11-depleted worms to varying doses of ionizing radiation. L4 worms were maintained in the presence or in the absence of 1 mM auxin for 24 hr, followed by irradiation at the indicated dosage, and then incubated for additional 8 hr to allow irradiated nuclei to progress to late pachynema. Scale bars, 5 μm.

DOI: https://doi.org/10.7554/eLife.30789.017

The following figure supplements are available for figure 5:

**Figure supplement 1.** Validation of protein depletion, and further characterization of ZHP-1/2.

DOI: https://doi.org/10.7554/eLife.30789.018

**Figure supplement 2.** Partial depletion fails to separate the roles of ZHP-1/2 in limiting late recombination intermediates and promoting crossover maturation.

DOI: https://doi.org/10.7554/eLife.30789.019

exposed these animals to varying doses of ionizing radiation. When SPO-11 alone was depleted by the AID system, GFP-COSA-1 foci and bivalents were eliminated. Following 10 Gy of radiation, a dose sufficient to ensure CO formation on most chromosomes in the absence of SPO-11, six bright GFP-COSA-1 foci were observed in each nucleus, as previously described (*Libuda et al., 2013*; *Yokoo et al., 2012*) (*Figure 5—figure supplement 1C*). When SPO-11 and ZHP-2 were co-depleted, bright GFP-COSA-1 foci were observed following low doses of radiation, but these never exceeded two per nucleus (*Figure 5D*). At 10 Gy, only extremely dim GFP-COSA-1 foci were detected (*Figure 5D*).

Taken together, these observations indicate that ZHP-1/2 are dispensable for a single break to become a functional interhomolog CO, but act to limit or focus the activity of ZHP-3/4 and/or other pro-CO factors to select a single recombination intermediate on each chromosome and to ensure CO maturation in the context of excess DSBs. A higher fraction of recombination intermediates become crossovers when breaks are severely reduced, but even when very few DSBs are present, ZHP-1/2 contribute to the accumulation of pro-CO factors, which in turn promotes CO maturation.

These results indicate that ZHP-1/2 play dual roles: they restrict the accumulation of pro-CO factors to a subset of recombination intermediates and promote their accumulation at a subset of sites. We tested whether these two functions could be separated by reducing the amount of either protein. When strains carrying AID-tagged ZHP-1 or -2 were exposed to low concentrations of auxin, these proteins could be partially depleted (*Figure 5—figure supplement 2A–B* and data not shown). At very low protein levels, we saw a reduction in the number of GFP-COSA-1 foci, and never observed more than six bright COSA-1 foci per nucleus. Thus, concentrations of ZHP-1/2 required to promote accumulation of pro-CO factors are also sufficient to limit their association with recombination sites (*Figure 5—figure supplement 2C* and data not shown), indicating that these functions are tightly coupled.

## ZHP-1/2 mediate chromosome remodeling in response to CO designation

As described in the Introduction, stepwise chromosome segregation in *C. elegans* meiosis requires functional differentiation of the chromosome regions on either side of the crossover. Here we report that even prior to any apparent reorganization of HTP-1/2 or SC proteins, ZHP-1/2 became restricted to the SC on one side of the CO, the presumptive short arm (*Figure 1* and data not shown). In addition, depletion of ZHP-1 or -2 resulted in aberrant chromosome remodeling: HTP-1/2 and SYP-1 persisted along most chromosomes, both recombinant and achiasmate (*Figure 6—figure supplement 1A–C*). These observations suggested that ZHP-1/2 might regulate chromosome remodeling, in addition to their roles in CO control.

To further probe the contribution of ZHP-1/2 to remodeling, we examined *dsb-2* mutants, in which some recombination intermediates accumulate pro-CO factors and undergo CO maturation even in the absence of ZHP-1/2. The low number of COs in *dsb-2* mutants also allows direct comparison between recombinant and nonexchange chromosomes in the same nucleus, and thus at the same cell cycle stage. As described above, at 48 h-post L4, most late pachytene nuclei in *dsb-2* mutants had either 0 or 1 COSA-1 focus. These foci also accumulated ZHP-3 and ZHP-4, which disappeared from the corresponding SCs on both sides of these sites (*Figure 6A* and data not shown). SC proteins and the Polo kinase PLK-2 also became enriched along these chromosomes relative to nonexchange chromosomes (*Figure 6A and D*), as previously described (*Machovina et al., 2016*; *Pattabiraman et al., 2017*). We also observed that PLK-2, which plays a poorly characterized role in remodeling (*Harper et al., 2011*), became co-enriched with ZHP-1/2 on one side of each CO (*Figure 6E*, upper panel). Similar enrichment of PLK-2 along the short arm was observed in wild-type animals (*Figure 6—figure supplement 2A–B*). This had not been apparent using PLK-2 antibodies, but was readily detected with epitope-tagged PLK-2, as recently reported (*Harper et al., 2011*; *Pattabiraman et al., 2017*).

When we co-depleted DSB-2 and ZHP-1/2, ZHP-3 still accumulated at the few bright COSA-1 foci and was depleted from the corresponding SCs on both sides of these sites (*Figure 6B*). This indicates that accumulation of pro-CO factors, rather than ZHP-1/2 activity per se, leads to depletion of ZHP-3/4 from the SC. However, none of the other hallmarks of CO-induced remodeling were observed in these animals, including accumulation of SC proteins and PLK-2, and asymmetrical localization of PLK-2, HTP-1/2, and LAB-1 (*Figure 6B–E* and data not shown). Instead, a small focus of PLK-2 was detected at CO sites (*Figure 6E*, lower panel). During diplotene-diakinesis, SC proteins also became restricted to this focus, while HTP-1/2 persisted along all chromosome arms (*Figure 6C*, lower panel).

We also depleted PLK-2 with the AID system (*Figure 6F*). Under these conditions, both the accumulation of pro-CO factors at a subset of recombination intermediates and the asymmetric localization of ZHP-1/2 were delayed, but still occurred (*Figure 6F*, lower panel). Thus, ZHP-1/2 are required for the asymmetric localization of PLK-2, but not vice versa, placing ZHP-1/2 upstream of PLK-2 in the remodeling pathway.

Because ZHP-3/4 are strictly required for CO formation, we could not analyze their roles in chromosome remodeling. However, prior work has found that a GFP-tagged ZHP-3 transgene acts as a separation-of-function allele that is proficient for CO formation but defective for remodeling (*Bhalla et al., 2008*).

This remodeling process ultimately leads to the spatial partitioning of cohesin complexes containing distinct kleisin subunits to reciprocal chromosome domains (*Severson et al., 2009*; *Severson and Meyer, 2014*). The most mature oocyte in each gonad arm, referred to as the '−1 oocyte,' normally shows enrichment of REC-8 along the long arms of each bivalent, while COH-3/4 are retained along the short arms. In ZHP-1/2 depleted worms, REC-8 persisted along all arms, while COH-3/4 virtually disappeared by the −1 oocyte (*Figure 6—figure supplement 2C*), consistent with the absence of other 'short arm' hallmarks.

Taken together, our analysis indicates that the ZHP circuit acts at or near the top of the hierarchy that mediates chromosome remodeling in response to CO designation, and directly couples these two key mechanisms to promote proper chromosome segregation. Our observations also imply that the persistence of SC along the short arm following CO designation plays an important role in remodeling by maintaining ZHP-1/2 and PLK-2 activity along this region. Together with previous observations, our findings indicate that a biochemical signaling cascade acts in *cis* within individual SC compartments to mediate and respond to CO designation (*Libuda et al., 2013*; *Machovina et al., 2016*).

## Compartmentalization of CO regulation by the synaptonemal complex

In *C. elegans* lacking the chromosome axis protein HTP-3, large polycomplexes appear shortly after meiotic entry and persist throughout meiotic prophase (*Goodyer et al., 2008*; *Rog et al., 2017*; *Severson and Meyer, 2014*). These polycomplexes recapitulate both the internal order and the liquid-like behavior of SCs, and ZHP-3 concentrates within these bodies (*Rog et al., 2017*). At mid-prophase, a focus of COSA-1 appears at the surface of each polycomplex. ZHP-3 also accumulates at these foci and gradually disappears from the interior of polycomplexes. Thus, ZHP-3 and COSA-1



**Figure 6.** ZHP-1/2 act at the top of a hierarchy of chromosome remodeling factors. (**A**) Late pachytene nuclei in a *dsb-2* mutant at 48 hr post-L4, stained for ZHP-3-V5 (green) and SYP-2 (red). ZHP-3 is depleted and SYP-2 is enriched specifically along SCs with bright ZHP-3 foci. Boxed areas indicate nuclei shown at higher magnification to the right. Arrowheads in inset images indicate designated CO sites. (**B**) Late pachytene nuclei stained for ZHP-3-V5 (green), SYP-2 (red) and GFP-COSA-1 (blue). In the absence of ZHP-2 and DSB-2, ZHP-3 was still depleted from CO-designated SCs.

*Figure 6 continued on next page*

*Figure 6 continued*

However, enrichment of SYP proteins was not observed, (**C**) Higher-magnification images of diplotene nuclei from hermaphrodites depleted of DSB-2 alone or in combination with ZHP-2. The tissue was stained with antibodies against SYP-1 (green), HTP-1 (red) and GFP-COSA-1 (blue). In the absence of DSB-2 alone, SYP-1 and HTP-1 localized to reciprocal chromosome domains of bivalent chromosomes, as previously described (*Machovina et al., 2016*; *Martinez-Perez et al., 2008*). In worms depleted of both ZHP-2 and DSB-2, SYP-1 was restricted to CO sites at diplotene-diakinesis, while HTP-1/2 were retained on both short and long arms of bivalents. Asterisk indicates nuclear envelope staining due to cross-reactivity of our SYP-1 antibody, which becomes more prominent in late prophase. (**D and E**) Representative nuclei from the same genotypes as in (**C**), stained for PLK-2-HA (green), anti-SYP-2 (red) and GFP-COSA-1 (blue). In the absence of DSB-2, PLK-2 became enriched on CO-designated chromosomes, after which it relocalized to the short arm of CO-designated SCs at late pachynema, as recently described (*Pattabiraman et al., 2017*). However, when ZHP-2 and DSB-2 were co-depleted, PLK-2 was not enriched along CO-designated chromosomes, although foci were observed at CO sites. (**F**) Representative late pachytene and diplotene nuclei from hermaphrodites depleted of PLK-2, stained for ZHP-2-HA (green), SYP-1 (red) and GFP-COSA-1 (blue). In the absence of PLK-2, CO designation and asymmetric localization of ZHP-2 were delayed, and asymmetric disassembly of the SC was severely impaired, as previously described (*Harper et al., 2011*), but ZHP-2 still relocalized to the presumptive short arm during late prophase. Arrowheads indicate designated CO sites. Scale bars, 5 μm.

DOI: https://doi.org/10.7554/eLife.30789.020

The following figure supplements are available for figure 6:

**Figure supplement 1.** ZHP-1/2 are required for chromosome remodeling.

DOI: https://doi.org/10.7554/eLife.30789.021

**Figure supplement 2.** ZHP-1/2 are required for chromosome remodeling.

DOI: https://doi.org/10.7554/eLife.30789.022

recapitulate the dynamic relocalization and interdependence that they show at CO sites on the surface of polycomplexes, even in the absence of DSBs or recombination intermediates (*Rog et al., 2017*).

To investigate this circuitry further, we disrupted *htp-3* in various tagged strains. All 4 ZHP proteins localized throughout polycomplexes during early prophase (*Figure 7A–B*, and data not shown), consistent with the idea that they associate with SC proteins. In contrast to ZHP-3 (*Rog et al., 2017*), ZHP-1/2 did not accumulate at COSA-1 foci, but instead remained diffusely localized throughout polycomplexes during late prophase (*Figure 7A–B* and data not shown). COSA-1 foci appeared during late prophase even when ZHP-1/2 were depleted, but under these conditions, ZHP-3 remained distributed throughout polycomplexes in addition to concentrating with COSA-1 (*Figure 7C*), further indicating that ZHP-1/2 promote the accumulation of ZHP-3/4. We also found that MSH-5 colocalized with ZHP-3/4 and COSA-1 foci on polycomplexes (*Figure 7D*). PLK-2 localized throughout polycomplexes even prior to the detection of COSA-1 foci, but became noticeably brighter afterwards, in the presence or absence of ZHP-1/2 (*Figure 7E–F*).

Together with the observations of *dsb-2* mutants described above, these findings demonstrate that key aspects of the regulatory circuit that designates a subset of recombination intermediates for CO formation act autonomously within each contiguous SC or polycomplex. Remarkably, our study indicates that this circuit is triggered within polycomplexes in response to progression of nuclei into late prophase, despite the absence of any DSBs or CO intermediates in *htp-3* mutants. These observations indicate that the 'designation' of one and only one site per compartment, which mirrors the logic of crossover assurance and interference, is an intrinsic property of this compartmentalized circuitry.

## Discussion

### Spatial compartmentalization of CO control

Evidence that the SC behaves as a phase-separated, liquid crystalline compartment suggested how this material might act as a conduit for diffusion of biochemical signals along the interface between paired chromosomes (*Rog et al., 2017*). Here we provide evidence that CO assurance, CO interference, and CO maturation are coordinately regulated by the ZHP RING finger proteins (*Figure 8*), all of which are concentrated within the SC. Spatial confinement of the ZHPs within this interface between homologs can explain how each pair of chromosomes in the same nucleus is regulated independently, although CO designation is normally coupled to global cell cycle signals. These

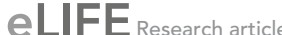

**Figure 7.** Compartmentalization of CO signaling within polycomplexes. Polycomplexes assemble from SC proteins in meiotic nuclei of *htp-3* mutants. During late meiotic prophase, a single focus containing COSA-1 and ZHP-3 appears on each polycomplex (***Rog et al., 2017*** and data not shown). (**A**) ZHP-1 (not shown) and ZHP-2 (green) localize throughout polycomplexes (red) both before and after the appearance of COSA-1 foci (blue). Insets show higher magnification images of the indicated polycomplexes without (left) and with (right) COSA-1 foci. (**B,C**) Late prophase nuclei showing polycomplexes in the presence and absence of ZHP-2 (minus and plus auxin treatment, respectively). Insets show higher magnification views of the indicated polycomplexes. (**B**) In the presence of ZHP-2, ZHP-3 (green) colocalizes with COSA-1 foci (cyan) in late meiotic prophase and becomes depleted from the body of polycomplexes (red). (**C**) In the absence of ZHP-2, ZHP-3 (green) colocalizes with GFP-COSA-1, but also remains diffusely localized throughout polycomplexes. (**D**) Late prophase nuclei from *htp-3* mutants showing colocalization of MSH-5 (red) with GFP-COSA-1 foci (green) on the surface of polycomplexes (blue). (**E,F**) Late prophase nuclei from the same genotypes depicted in (**B,C**). (**E**) PLK-2 (green) localizes throughout

*Figure 7 continued on next page*

*Figure 7 continued*

polycomplexes (red) prior to the appearance of GFP-COSA-1 foci (blue), but staining becomes more intense in late prophase. (**F**) Depletion of ZHP-2 does not affect the localization of PLK-2 to polycomplexes. Scale bars in lower magnification images, 5 µm; in insets, 1 µm.

DOI: https://doi.org/10.7554/eLife.30789.023

results add further support to the idea that CO regulation is mediated through this liquid crystalline medium.

CO interference in *C. elegans* normally limits COs to one per homolog pair. This 'winner takes all' logic is reminiscent of other well-studied biological symmetry-breaking circuits, such as the selection of a single bud site by growing *S. cerevisiae* cells (*Goryachev and Pokhilko, 2008*; *Howell et al., 2012*; *Ozbudak et al., 2005*). Our evidence indicates that ZHP-1/2 are crucial for both positive feedback that promotes accumulation of pro-CO factors, including ZHP-3/4, at a single recombination intermediate, and negative feedback that depletes these factors from other candidate sites

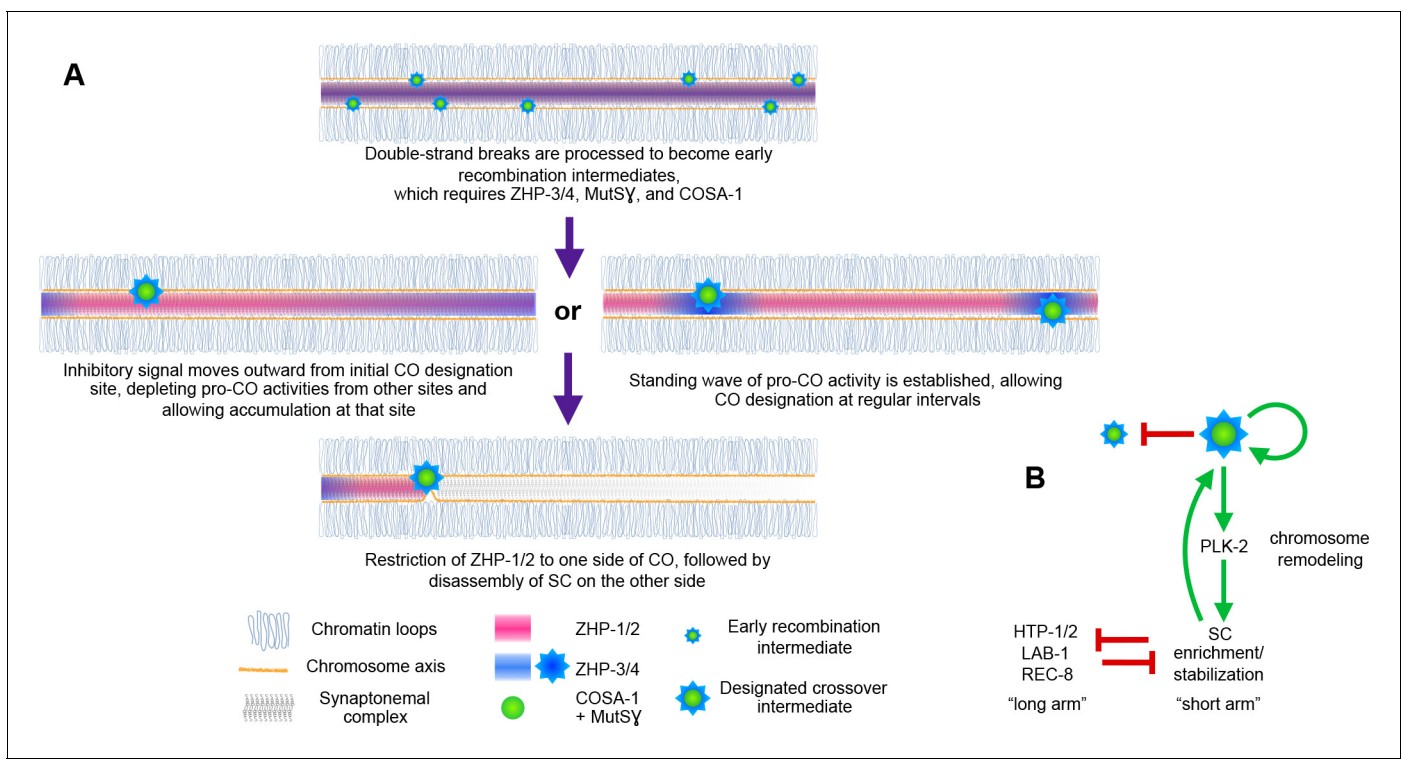

**Figure 8.** Schemata for crossover control and chromosome remodeling. (**A**) Spatial regulation of CO designation by ZHP proteins. In early prophase, all 4 ZHPs concentrate within SCs at the interface between homologous chromosomes. DSBs are induced and homologous recombination is initiated through strand invasion. Recruitment or stabilization of MutSγ and COSA-1 at these early intermediates depends on the presence of ZHP-3/4 within the SC. As nuclei progress through mid-prophase, one such intermediate along each chromosome stochastically crosses a threshold activity level. [We note that this is analogous to a 'cracking' event posited in a beam-film model for crossover interference (*Kleckner et al., 2004*; *Zhang et al., 2014b*)]. This event triggers positive and negative feedback that promotes accumulation of COSA-1, MutSγ, and ZHP-3/4 at this site. These pro-CO activities may be depleted from nearby recombination intermediates through a wave of inhibitory activity that travels outward from a designated CO site; alternatively, local activation and lateral inhibition may give rise to a periodic pattern of pro-CO activity along each pair of chromosomes. In either scenario, diffusion of proteins within the SC permits coordinated behavior of potential CO sites along each pair of chromosomes. (**B**) A circuit diagram for CO control and chromosome remodeling, incorporating the findings of this work. Coupled positive and negative feedback mechanisms mediated by ZHPs and other factors lead to accumulation of pro-CO factors at a single designated crossover site. ZHP-1/2 also act at the top of a hierarchy that leads to PLK-2 enrichment on one side of the crossover and retention of the SC within this domain, creating another positive feedback loop, although how ZHP-1/2 become enriched on one side of the crossover is currently unknown. Activities within the SC antagonize the retention of HTP-1/2 and associated factors along the axis, and vice versa, resulting in the differentiation of two distinct chromosome domains. This leads to eventual removal of REC-8 from the short arm and its retention on the long arm, promoting accurate, stepwise meiotic chromosome separation.

DOI: https://doi.org/10.7554/eLife.30789.024

(*Figure 8*). Similar feedback loops have been proposed in previous work on crossover regulation (*Qiao et al., 2014*; *Rao et al., 2017*; *Reynolds et al., 2013*; *Yokoo et al., 2012*). Indeed, coupled positive and negative feedback mechanisms are likely essential for any robust, switch-like decision executed by biological circuits (*Pfeuty and Kaneko, 2009*). A full understanding of CO control will clearly require identification of the substrates of the ZHP ligase family and the effects of their modification.

A single crossover site could be selected, and other sites deselected, via a 'trigger wave' that moves rapidly through the SC from an initial site of stochastic CO designation and inhibits selection of nearby recombination intermediates (*Figure 8A*). However, such a mechanism cannot easily be reconciled with observations that pro-crossover factors, including Zip3, accumulate at recombination intermediates even prior to completion of synapsis in budding yeast, and likely in other organisms. Alternatively, coupled positive and negative regulation may give rise to a periodic standing wave, or Turing pattern (*Turing, 1952*), of CO designation potential along each chromosome. This type of patterning requires interaction between two diffusible activities: an activating activity – or 'morphogen,' to use Turing's term – that acts locally and shows autocatalytic or cooperative behavior, and an inhibitory activity that acts over a greater distance. This type of pattern has intrinsic periodicity that depends on the activities and diffusion rates of the activator and inhibitor. Such a pattern could thus be established prior to full SC assembly, in which case designation of the first CO site would spatially fix one node of the pattern (*Figure 8A*). A Turing patterning mechanism is also consistent with evidence presented here and in previous studies that stabilization of recombination intermediates and CO patterning are tightly coupled. Evidence that mammalian RNF212 and Arabidopsis HEI10 act as dosage-sensitive modulators of recombination rates (*Qiao et al., 2014*; *Reynolds et al., 2013*; *Ziolkowski et al., 2017*) is compatible with either a temporally-dependent or standing wave model, since an increase or decrease in these activities could alter either the rate of signal propagation or the frequency of a waveform along paired chromosomes.

While the SC is an unusual, liquid crystalline material, liquid-liquid phase separation (LLPS) has been implicated in subcellular compartmentalization of diverse signaling mechanisms (*Banani et al., 2017*). By diffusing and interacting within a spatially confined liquid compartment, biochemical signals could potentially establish a variety of periodic, Turing-type patterns within cells. Analogously, diffusion of signaling molecules within cell membranes can create two-dimensional patterns, which have been proposed to orchestrate polarity decisions (*Goryachev and Pokhilko, 2008*), as well as more elaborate patterning such as hexagonal arrays of stereocilia (*Jacobo and Hudspeth, 2014*).

Intriguingly, the ZHP proteins and their homologs are very similar to TRIM E3 ligases (tripartite motif: RING, B-box, coiled-coil domain), a very large family of enzymes with diverse cellular roles. The functions of most TRIM ligases are unknown, but like the ZHP proteins, they often form homo- or heterodimers, and many of these proteins appear to localize to cellular compartments, including globular nuclear and cytoplasmic bodies, or to filamentous structures such as microtubules (*Rajsbaum et al., 2014*; *Reymond et al., 2001*). Thus, we speculate that compartmentalized regulatory mechanisms with similarities to the circuitry illuminated here may control other spatially-patterned activities within cells.

## Conservation and divergence of the Zip3/ZHP protein family

RING finger proteins homologous to the ZHP family play important roles in CO formation in diverse eukaryotes (*Agarwal and Roeder, 2000*; *Ahuja et al., 2017*; *Chelysheva et al., 2012*; *Gray and Cohen, 2016*; *Lake et al., 2015*; *Qiao et al., 2014*; *Rao et al., 2017*; *Reynolds et al., 2013*; *Wang et al., 2012*; *Ward et al., 2007*; *Zhang et al., 2014a*). Interestingly, the number of these proteins varies among organisms. The cytological distributions and apparent functions of ZHP-3/4 and ZHP-1/2 are very similar to those reported for the mammalian recombination regulators RNF212 and HEI10, respectively. Like ZHP-3/4, mouse RNF212 is required to stabilize recombination intermediates (*Reynolds et al., 2013*), while HEI10, like ZHP-1/2, is required to limit the number of RNF212/ MutSγ-positive foci along chromosomes and for robust CO maturation (*Qiao et al., 2014*). Most ZHP homologs also contain predicted coiled-coil domains and localize to the central region of the SC, including Zip3 in budding yeast, which forms foci along chromosomes but localizes throughout polycomplexes. A notable exception is a recently identified homolog required for crossovers in the ciliate *Tetrahymena thermophila*. This protein lacks a coiled-coil domain, likely related to the fact that this organism lacks SCs (*Shodhan et al., 2017*).

HEI10 in *Sordaria* and *Arabidopsis*, and Zip3 in budding yeast, may be bifunctional proteins with roles similar to both ZHP-1/2 and ZHP-3/4, as suggested by their localization to CO sites and the apparent absence of RNF212 in these clades (*Agarwal and Roeder, 2000*; *Chelysheva et al., 2012*; *De Muyt et al., 2014*). We speculate that they could play dual roles by partnering with multiple E2 conjugating enzymes, or perhaps with distinct substrate-recognition factors.

In mammals, RNF212 and HEI10 are thought to mediate opposing pro- and anti-crossover activities through SUMOylation and SUMO-dependent ubiquitylation, respectively (*Qiao et al., 2014*; *Rao et al., 2017*; *Reynolds et al., 2013*); a similar 'SUMO-ubiquitin switch' has been proposed based on findings in Sordaria (*De Muyt et al., 2014*). In budding yeast, Sordaria, and mice, SUMOylation is essential for SC assembly, and thus for CO regulation (*De Muyt et al., 2014*; *Hooker and Roeder, 2006*; *Klug et al., 2013*; *Leung et al., 2015*; *Lin et al., 2010*; *Rao et al., 2017*; *Voelkel-Meiman et al., 2013*). In *C. elegans* we do not detect SUMO cytologically either along the SC or at CO sites, even with highly sensitive tools such as epitope-tagged SUMO transgenes (*Pelisch and Hay, 2016*; data not shown). Further, genetic analysis has indicated that SMO-1 (SUMO) and its E2 conjugating enzyme UBC-9 are dispensable for synapsis and CO control in *C. elegans*, although they play key roles downstream of COs to mediate meiotic chromosome segregation in oocytes (*Bhalla et al., 2008*; *Pelisch et al., 2017*). Taken together, we favor the idea that each pair of ZHP proteins acts as a ubiquitin ligase, like the 'lion's share' of RING finger proteins (*Deshaies and Joazeiro, 2009*).

Our analysis also reveals that the *C. elegans* ZHP proteins likely function as heterodimers, or perhaps as higher order oligomers. It will be interesting to determine whether ZHP homologs in other organisms also have obligate partners, or perhaps homooligomerize, since this knowledge may help to illuminate their in vivo activities and substrates.

## Regulation of sister chromatid cohesion by the ZHP family

In addition to their roles in CO control, the ZHP family also link COs to chromosome remodeling, which enables the stepwise release of cohesion during the MI and MII divisions (*Figure 8*). Specifically, we find that CO designation results in enrichment of ZHP-1/2 along one chromosome arm, which in turn promotes concentration of PLK-2 within this domain, asymmetric disassembly of the SC, and spatial regulation of cohesin function.

It is not clear whether sister chromatid cohesion during meiosis is regulated in the same fashion in other organisms. In budding and fission yeasts, Drosophila, diverse plants, and mammals, cohesion near centromeres is specifically retained during the first division (*Duro and Marston, 2015*). Cohesin complexes containing the meiosis-specific kleisin Rec8 are enriched in pericentromeric heterochromatin and sometimes at centromere cores, and Shugoshin/MEI-S332, a Rec8-specific protective factor, is also recruited to these regions. In principle, programmed release of cohesion along arms and its retention near centromeres could suffice to allow homologs to segregate in MI while maintaining a link between sisters until MII. However, experimental evidence from several organisms indicates that CO formation leads to a local disruption of cohesion, which can predispose chromosomes to missegregate when COs occur close to centromeres (*Brar and Amon, 2008*). In light of other similarities between ZHP-1/2 and HEI10, this raises the possibility that HEI10 may also influence sister chromatid cohesion, perhaps via spatial control of Polo-like kinases, which promote release of cohesion in mitosis and meiosis (*Archambault and Glover, 2009*). If so, HEI10 activity may directly impact chromosome missegregation in human oocytes, which has been attributed to compromised cohesion, and also – very recently – to inefficient CO maturation (*Wang et al., 2017*), another process that involves HEI10.

## Materials and methods

### Key resources table

| Reagent type (species) or resource | Designation | Source or reference | Identifiers | Additional information |
|---|---|---|---|---|
| Gene (*Caenorhabditis elegans*) | *zhp-1* | WormBase/This paper | WormBase ID: F55A12.10; WBGene00018867 | |

*Continued on next page*

*Continued*

| Reagent type (species) or resource | Designation | Source or reference | Identifiers | Additional information |
|---|---|---|---|---|
| Gene (*Caenorhabditis elegans*) | *zhp-2* | WormBase/This paper | WormBase ID: D1081.9; WBGene00008387 | |
| Gene (*Caenorhabditis elegans*) | *zhp-3* | WormBase/ (*Jantsch et al., 2004*) PMID: 15340062 | WormBase ID: K02B12.8b; WBGene00006976 | |
| Gene (*Caenorhabditis elegans*) | *zhp-4* | WormBase/This paper | WormBase ID: Y39B6A.16; WBGene00012678 | |
| Strain, strain background (*Saccharomyces cerevisiae*) | *S. cerevisiae*: MaV203 | Invitrogen, Thermo Fisher Scientific, Waltham, MA | Cat#11445012 | |
| Strain, strain background (*Caenorhabditis elegans*) | For *C. elegans* allele and strain information, see **Supplementary file 1**, Tables S2 and S4. | This paper | N/A | |
| Genetic reagent | For CRISPR/Cas9 reagents, see sequence-based reagent and peptide, recombinant protein. | This paper | N/A | |
| Antibody | Rabbit polyclonal anti-SYP-1 | (*MacQueen et al., 2002*) PMID: 12231631 | N/A | (1:500 IF) |
| Antibody | Rabbit polyclonal anti-SYP-2 | (*Colaiácovo et al., 2003*) PMID: 12967565 | N/A | (1:500 IF) |
| Antibody | Rabbit polyclonal anti-HTP-1 | (*Martinez-Perez et al., 2008*) PMID: 18923085 | N/A | (1:500 IF) |
| Antibody | Rabbit polyclonal anti-RAD-51 | Millipore Sigma, St. Louis, MO | Cat#29480002 | (1:5,000 IF) |
| Antibody | Rabbit polyclonal anti-pHIM-8/ZIMs | (*Kim et al., 2015*) PMID: 26506311 | N/A | (1:500 IF) |
| Antibody | Rabbit polyclonal anti-ZHP-3 | ModENCODE/SDIX; (*Rog et al., 2017*) PMID: 28045371 | Cat#SDQ3956 | (1:5,000 IF) |
| Antibody | Rabbit polyclonal anti-MSH-5 | ModENCODE/SDIX | Cat#SDQ2376 | (1:5,000 IF) |
| Antibody | Rabbit polyclonal anti-REC-8 | ModENCODE/SDIX | Cat#SDQ0802 | (1:5,000 IF) |
| Antibody | Rabbit polyclonal anti-COH-3 | ModENCODE/SDIX | Cat#SDQ3972 | (1:5,000 IF) |
| Antibody | Rat polyclonal anti-HIM-8 | (*Phillips et al., 2005*) PMID: 16360035 | N/A | (1:500 IF) |
| Antibody | Goat polyclonal anti-SYP-1 | (*Harper et al., 2011*) PMID: 22018922 | N/A | (1:300 IF) |
| Antibody | Chicken polyclonal anti-HTP-3 | (*MacQueen et al., 2005*) PMID: 16360034 | N/A | (1:500 IF) |
| Antibody | Guinea pig polyclonal anti-HTP-3 | (*MacQueen et al., 2005*) PMID: 16360034 | N/A | (1:500 IF) (1:1,500 WB) |
| Antibody | Guinea pig polyclonal anti-PLK-2 | (*Harper et al., 2011*) PMID: 22018922 | N/A | (1:100 IF) |
| Antibody | Rabbit polyclonal anti-V5 | Millipore Sigma | Cat#V8137; RRID:AB_261889 | (1:250 IF) |
| Antibody | Mouse monoclonal anti-V5 | Thermo Fisher Scientific | Cat#R960-25; RRID:AB_2556564 | (1:500 IF) (1:1,000 WB) |
| Antibody | Mouse monoclonal anti-FLAG M2 | Millipore Sigma | Cat#F1804; RRID:AB_262044 | (1:500 IF) (1:1,000 WB) |

*Continued on next page*

*Continued*

| Reagent type (species) or resource | Designation | Source or reference | Identifiers | Additional information |
|---|---|---|---|---|
| Antibody | Mouse monoclonal anti-HA, clone 2–2.2.14 | Thermo Fisher Scientific | Cat#26183; RRID:AB_10978021 | (1:400 IF) (1:1,000 WB) |
| Antibody | Mouse monoclonal anti-GFP | Millipore Sigma | Cat#11814460001; RRID:AB_390913 | (1:500 IF) (1:1,000 WB) |
| Antibody | Mouse monoclonal anti-α-tubulin, clone DM1A | Millipore Sigma | Cat#05–829; RRID:AB_310035 | (1:5,000 WB) |
| Recombinant DNA reagent | $P_{eft-3}$::AID::GFP::*unc-54* 3'UTR | (*Zhang et al., 2015*) PMID: 26552885; Addgene | pLZ29; Addgene plasmid #71719 | |
| Recombinant DNA reagent | cDNA *zhp-1* | WormBase | WormBase ID: F55A12.10; WBGene00018867 | |
| Recombinant DNA reagent | cDNA *zhp-2* | WormBase | WormBase ID: D1081.9; WBGene00008387 | |
| Recombinant DNA reagent | cDNA *zhp-3* | WormBase | WormBase ID: K02B12.8b; WBGene00006976 | |
| Recombinant DNA reagent | cDNA *zhp-4* | WormBase (codon-optimized by Integrated DNA Technologies to facilitate gBlock synthesis) | WormBase ID: Y39B6A.16; WBGene00012678 | |
| Recombinant DNA reagent | Plasmid: pDEST32-*zhp-1*-V5 | This paper | N/A | |
| Recombinant DNA reagent | Plasmid: pDEST32-*zhp-2*-3xFLAG | This paper | N/A | |
| Recombinant DNA reagent | Plasmid: pDEST32-*zhp-3*-3xFLAG | This paper | N/A | |
| Recombinant DNA reagent | Plasmid: pDEST32-*zhp-4*-HA | This paper | N/A | |
| Recombinant DNA reagent | Plasmid: pDEST22-*zhp-1*-V5 | This paper | N/A | |
| Recombinant DNA reagent | Plasmid: pDEST22-*zhp-2*-3xFLAG | This paper | N/A | |
| Recombinant DNA reagent | Plasmid: pDEST22-*zhp-3*-3xFLAG | This paper | N/A | |
| Recombinant DNA reagent | Plasmid: pDEST22-*zhp-4*-HA | This paper | N/A | |
| Sequence-based reagent | FISH probe to the right arm of Chromosome V (5S rDNA) | (*Dernburg et al., 1998*) PMID: 9708740 | N/A | |
| Sequence-based reagent | FISH Probe to the right arm of X Chromosome | (*Phillips et al., 2005*) PMID: 16360035 | N/A | |
| Sequence-based reagent | CRISPR tracrRNA | Integrated DNA Technologies, Skokie, IL | Cat#1072534 | |
| Sequence-based reagent | For crRNAs, repair templates and genotyping primers, see *Supplementary file 1*, Table S3 | This paper | N/A | |
| Peptide, recombinant protein | *S. pyogenes* Cas9-NLS purified protein | QB3 MacroLab at UC Berkeley | N/A | |
| Chemical compound, drug | Auxin: 1H-Indole-3-acetic acid | Thermo Fisher Scientific, Waltham, MA | Cat#122160250; CAS:87-51-4 | |
| Chemical compound, drug | NuSieve 3:1 agarose | Lonza, Walkersville, MD | Cat#50090 | |
| Chemical compound, drug | Nuclease-Free Duplex Buffer | Integrated DNA Technologies | Cat#11-01-03-01 | |
| Commercial assay or kit | SuperSignal West Femto Maximum Sensitivity Substrate kit | Thermo Fisher Scientific | Cat #34095 | |

*Continued on next page*

*Continued*

| Reagent type (species) or resource | Designation | Source or reference | Identifiers | Additional information |
|---|---|---|---|---|
| Software, algorithm | SoftWorx package | Applied Precision; GE Healthcare Bio-Sciences, Pittsburgh, PA | http://www.gelifesciences.com/webapp/wcs/stores/servlet/productById/en/GELifeSciences-us/29065728 | |
| Software, algorithm | ImageJ | NIH | https://imagej.nih.gov/ij/ | |
| Software, algorithm | Adobe Photoshop CC 2014 | Adobe Systems, San Jose, CA | http://www.adobe.com/products/photoshop.html | |
| Software, algorithm | T-COFFEE | Swiss Institute of Bioinformatics; (*Notredame et al., 2000*) PMID: 10964570 | http://tcoffee.vital-it.ch/apps/tcoffee/index.html | |
| Software, algorithm | IBS_1.0.1 | (*Liu et al., 2015*) PMID: 26069263 | http://ibs.biocuckoo.org/download.php | |
| Software, algorithm | Clustal Omega | EMBL-EBI | http://www.clustal.org/omega/ | |
| Software, algorithm | Graphpad Prism | Graphpad Software, Inc., La Jolla, CA | http://www.graphpad.com/ | |
| Software, algorithm | MSG software package | (*Andolfatto et al., 2011*) PMID: 21233398 | https://github.com/Janelia SciComp/msg | |

## Generation of transgenic worm strains

All *C. elegans* strains were maintained on nematode growth medium (NGM) plates seeded with OP50 bacteria at 20°C. All epitope-tagged genes used in this study were fully functional, as indicated by their ability to support normal meiosis (*Supplementary file 1*, Table S1). New alleles and their detailed information are listed in *Supplementary file 1*, Tables S2 and S3. Unless otherwise indicated, new alleles in this study were generated by genome editing, by injection of Cas9-gRNA RNP complexes, using *dpy-10* Co-CRISPR to enrich for edited progeny (*Arribere et al., 2014*; *Paix et al., 2015*). Cas9-NLS protein was produced by the MacroLab at UC Berkeley. The protein was complexed *in vitro* with equimolar quantities of duplexed tracrRNA and target-specific crRNAs, purchased from Integrated DNA Technologies (IDT). Repair templates were single- or double-stranded amplicons generated by asymmetric or standard PCR, respectively, or Ultramer oligonucleotides purchased from IDT. Each 10 µl of injection mixture contained: 16 µM [Cas9 protein + gRNA] (target gene + *dpy-10*), plus 0.16–2 µM long and/or 6 µM oligonucleotide repair templates. In cases where Unc worms were injected (*e.g.,* to tag *zhp* genes in a *syp-1(me17)/nT1* strain), plasmids encoding green fluorescent reporter proteins were co-injected to enrich for edited progeny.

Injected animals were transferred to individual plates and incubated at 20 °C. After 4 days, F1 Rol or Dpy progeny (*dpy-10* heterozygotes or homozygotes, respectively) from 'jackpot' broods containing multiple Rol/Dpy animals were picked to individual plates, allowed to produce self-progeny, then lysed and screened by PCR.

Disruption of the *zhp-1* gene was accomplished by inserting the sequence TAAGCTCGAG after the 6th codon, introducing a stop codon, a frameshift, and an Xho I restriction site. Three independent alleles of the same sequence were generated: *ie43* and *ie44* in the N2 strain background, and *ie46* in the CB4856 strain. The *zhp-2* gene was similarly disrupted by insertion of TAATAATTAATTAG after the 7th codon, resulting in multiple stop codons and a frameshift.

See *Supplementary file 1*, Table S4 for a full list of strains used. Unless otherwise indicated, young adults were used for both immunofluorescence and Western blotting assays. Synchronized young adults were obtained by picking L4 larvae and culturing them for 20–24 hr at 20°C.

## Auxin-mediated protein depletion

To deplete AID-tagged proteins in a strain expressing the *AtTIR1* transgene, animals were treated with auxin (indole acetic acid, IAA) as previously described (*Zhang et al., 2015*). Briefly, NGM agar was supplemented with 1 mM IAA just before pouring plates. *E. coli* OP50 bacteria cultures were concentrated and spread on plates, which were allowed to dry before transferring *C. elegans* onto the plates.

We note that several major advantages of the AID system have facilitated our study of these proteins in meiosis: (i) this approach enables temporal control and germline-specific degradation of proteins; (ii) complex strains that combine various mutations and epitope-tagged alleles can be constructed far more easily, since balancer chromosomes are not required to maintain conditional alleles of genes essential for reproduction. Additionally, control experiments can be performed using the same strains without auxin treatment, and partial protein depletion can be achieved by titrating auxin concentrations.

## Viability and fertility

To quantify brood size and males among self-progeny, L4 hermaphrodites were picked onto individual plates and transferred to new plates daily over 4 days. Eggs were counted every day. Viable progeny and males were counted when the F1 reached the L4 or adult stages. To analyze viability and fertility in the presence of auxin, L1s were transferred onto auxin plates and grown to the L4 stage. Hermaphrodites were then transferred onto fresh auxin plates, transferred daily over 4 days, and progeny were scored as described above.

## Recombination mapping by whole-genome sequencing

The *zhp-1* gene was disrupted by CRISPR/Cas9-based genome editing in both the N2 Bristol and CB4856 Hawaiian strain backgrounds, as described above (allele designations: *ie43* and *ie46*, respectively). By crossing animals carrying these mutations, we generated N2/CB4856 hybrids lacking *zhp-1.* To map meiotic COs that occurred during oogenesis in these animals, hybrid hermaphrodites were backcrossed to (*zhp-1*$^+$) CB4856 males. Hermaphrodite progeny from this cross were picked to individual 60 mm plates and allowed to produce self-progeny until the plates were starved. Genomic DNA was extracted from these pools of progeny using Gentra Puregene Tissue Kit (QIAGEN, Hilden, Germany), and quantified by Qubit (Invitrogen). Sequencing libraries were constructed in the Functional Genomics Lab (FGL), a QB3-Berkeley Core Research Facility at UC Berkeley. DNA was Fragmented with an S220 Focused-Ultrasonicator (Covaris, Woburn, MA), then cleaned and concentrated with the MinElute PCR Purification kit (QIAGEN). Library preparation was done on an Apollo 324 with PrepX ILM 32i DNA Library Kits (WaferGen Biosystems, Fremont, CA). 9 cycles of PCR amplification were used. The average insert size of libraries was 380 bp. 150PE sequencing at approximately 9x coverage was carried out on an Illumina Hiseq 2500 by the Vincent J. Coates Genomics Sequencing Laboratory at UC Berkeley. Two libraries for each of the parental strains (N2 and CB4856) were prepared in parallel, sequenced at 30x coverage, and aligned to the reference genome sequences available through Wormbase or (*Thompson et al., 2015*). To map COs, the data were analyzed using the MSG software package (https://github.com/JaneliaSciComp/msg) (*Andolfatto et al., 2011*).

## Yeast two-hybrid analysis

To test for interactions among the ZHP proteins using the yeast two-hybrid system, coding sequences for ZHP-1-V5, ZHP-2-3xFLAG, ZHP-3-3xFLAG and ZHP-4-HA were synthesized as gBlocks by Integrated DNA Technologies (IDT). Epitope tags were appended to the protein sequences so that expression in yeast could be monitored. The coding sequence for ZHP-4 was codon-optimized by IDT to facilitate gBlock synthesis and expression in yeast. Each of these sequences was fused to both the GAL4 DNA binding domain and the GAL4 activation domain by insertion into pDEST32 and pDEST22, respectively, using the Gateway cloning system, following the manufacturer's instructions (Invitrogen).

All possible pairwise combinations of the four ZHP proteins, including potential homodimers, were tested using the ProQuest™ Two-Hybrid System (Invitrogen). Briefly, each pair of bait and prey plasmids was co-transformed into yeast strain MaV203 using standard LiAc-mediated transformation. To test for specific interactions, 2–4 independent clones of each strain were assayed for *HIS3*, *URA3* and *LacZ* induction. Negative and positive controls provided with the ProQuest™ kit (Cat. Series 10835) were run in parallel.

## Microscopy

Immunofluorescence experiments were performed as previously described (*Zhang et al., 2015*), except that samples were mounted in Prolong Diamond mounting medium with DAPI (Millipore Sigma). Primary antibodies were obtained from commercial sources or have been previously described, and were diluted as follows: Rabbit anti-SYP-1 [1:500, (*MacQueen et al., 2002*), Rabbit anti-SYP-2 [1:500, (*Colaiácovo et al., 2003*), Rabbit anti-HTP-1 [1:500, (*Martinez-Perez et al., 2008*), Rabbit anti-RAD-51 (1:5,000, Novus Biologicals, #29480002), Rabbit anti-pHIM-8/ZIMs [1:500, (*Kim et al., 2015*), Rabbit MSH-5 [1:5,000, modENCODE/SDIX, #SDQ2376, (*Yokoo et al., 2012*), Rabbit anti-REC-8 (1:5,000, modENCODE/SDIX, #SDQ0802), Rabbit anti-COH-3 (1:5,000, modENCODE/SDIX, #SDQ3972), Guinea pig anti-PLK-2 [1:100, (*Harper et al., 2011*), Goat anti-SYP-1 [1:300, (*Harper et al., 2011*), Chicken anti-HTP-3 [1:500, (*MacQueen et al., 2005*), Rabbit anti-ZHP-3 [1:5,000, modENCODE/SDIX, #SDQ3956, (*Rog et al., 2017*), Rat anti-HIM-8 [1:500, (*Phillips et al., 2005*), Mouse anti-FLAG (1:500, Sigma, #F1804), Mouse anti-HA (1:400, Thermo Fisher, #26183), Mouse anti-V5 (1:500, Thermo Fisher, #R960-25), Rabbit anti-V5 (1:250, Millipore Sigma, #V8137), Mouse anti-GFP (1:500, Millipore Sigma, #11814460001). Secondary antibodies labeled with Alexa 488, Cy3 or Cy5 were purchased from Jackson ImmunoResearch (West Grove, PA) and used at 1:500.

Chromosomal *in situ* hybridization was performed as previously described (*Dernburg et al., 1998*) with modifications. Briefly, worms were dissected and fixed essentially according to the immunofluorescence protocol described above. Fixed worm gonads were then washed with 2xSSCT and incubated in 50% formamide/2xSSCT overnight at 37°C. Tissue and probes were denatured at 91°C for 2 min, then hybridized overnight at 37 °C. Hybridized gonads were then washed and stained with DAPI. Two chromosome-specific probes were used: an oligonucleotide targeting a short repetitive sequence on the X chromosome (*Lieb et al., 2000*; *Phillips et al., 2005*) and the 5S rDNA repeat on the right arm of chromosome V (*Dernburg et al., 1998*). FISH probes were labeled with aminoallyl-dUTP (Millipore Sigma) by terminal transferase-mediated 3' end-labeling, followed by conjugation to Alexa 488-NHS-ester (Thermo Fisher Scientific) or Cy3-NHS-ester (Themo Fisher Scientific), as previously described (*Dernburg, 2011*)

All images were acquired as z-stacks through 8 µm depth at intervals of 0.2 µm using a DeltaVision Elite microscope (GE) with a 100x, 1.4 N.A. oil-immersion objective. Iterative 3D deconvolution, image projection and colorization were carried out using the SoftWoRx package and Adobe Photoshop CC 2014.

## Western blotting

Adult worms were picked into SDS sample buffer and lysed by boiling for 30 min, a single freeze/thaw cycle in liquid nitrogen, and occasionally vortexing. Whole worm lysates were separated on 4–12% polyacrylamide gradient gels and blotted with indicated antibodies: Mouse anti-GFP (1:1,000, Millipore Sigma, #11814460001), Mouse anti-FLAG (1:1,000, Millipore Sigma, #F1804), Mouse anti-HA (1:1,000, Thermo Fisher, #26183), Mouse anti-V5 (1:1,000, Thermo Fisher, #R960-25), Mouse anti-α-tubulin (1:5,000, Millipore Sigma, #05–829), Guinea pig anti-HTP-3 [1:1,500, (*MacQueen et al., 2005*). HRP-conjugated secondary antibodies (Jackson Immunoresearch #115-035-068) and SuperSignal West Femto Maximum Sensitivity Substrate (Thermo Fisher, #34095) were used for detection.

To quantify Western blots, TIF images were collected for each blot using a Chemidoc system (Bio-Rad, Hercules, CA). Integrated intensities for relevant bands were then calculated using ImageJ. To normalize for sample loading, the indicated band intensity was divided by the corresponding α-tubulin or HTP-3 band intensity. Each normalized band intensity is expressed as the percentage of the intensity at t = 0 or in a control sample.

## Quantification and statistical analysis

Quantification methods and statistical parameters are indicated in the legend of each figure, including error calculations (SD or SEM), *n* values, statistical tests, and p-values. p<0.05 was considered to be significant.

## Acknowledgements

This work was supported by funding from the National Institutes of Health (R01 GM065591) and the Howard Hughes Medical Institute to AFD. Some strains were provided by the CGC, which is funded by NIH Office of Research Infrastructure Programs (P40 OD010440). Genome sequencing was performed in the Vincent J Coates Genomics Sequencing Laboratory at UC Berkeley, supported by NIH S10 Instrumentation Grants S10RR029668 and S10RR027303. We are grateful to Ed Ralston for assistance with the irradiator and to Anne Villeneuve for the SYP-2 antiserum. We thank members of the Dernburg lab for helpful discussions during the course of this work and for critical reading of the manuscript, Mike Rosen for pointing us to work on TRIM ligases, Leonid Mirny for calling our attention to the pioneering work of Alan Turning, Federico Pelisch for helpful comments on the manuscript, and anonymous reviewers for their constructive suggestions.

## Additional information

### Competing interests

Abby F Dernburg: Reviewing editor, *eLife*. The other authors declare that no competing interests exist.

### Funding

| Funder | Grant reference number | Author |
| --- | --- | --- |
| Howard Hughes Medical Institute | | Liangyu Zhang<br>Regina Rillo-Bohn<br>Abby F Dernburg |
| National Institute of General Medical Sciences | GM065591 | Abby F Dernburg |
| National Institutes of Health | P40 OD010440 | Abby F Dernburg |

The funders had no role in study design, data collection and interpretation, or the decision to submit the work for publication.

### Author contributions

Liangyu Zhang, Conceptualization, Data curation, Formal analysis, Investigation, Writing—original draft, Writing—review and editing; Simone Köhler, Formal analysis, Methodology, Writing—review and editing; Regina Rillo-Bohn, Investigation, Writing—review and editing; Abby F Dernburg, Conceptualization, Funding acquisition, Investigation, Writing—original draft, Project administration, Writing—review and editing

### Author ORCIDs

Liangyu Zhang http://orcid.org/0000-0002-2701-0773
Abby F Dernburg http://orcid.org/0000-0001-8037-1079

### Decision letter and Author response

Decision letter https://doi.org/10.7554/eLife.30789.030
Author response https://doi.org/10.7554/eLife.30789.031

## Additional files

### Supplementary files

• Supplementary file 1. This file includes four tables (Table S1-S4). Table S1 reports analysis of the progeny produced by animals expressing epitope-tagged proteins described in this work. In all cases, animals homozygous for these epitope-tagged alleles show no reduction in embryonic viability or increase in male production, indicating that the epitope tags used to detect various proteins do not impair their meiotic functions. Table S2 provides a list of all new alleles generated in this

work, with details about the genome editing methods used to introduce epitope tags or mutations. Table S3 includes the sequences of RNA and DNA sequences used for genome editing, including synthetic CRISPR RNAs (crRNAs), repair templates, and DNA primers used for detecting or genotyping these mutations. In cases where restriction digests were used for genotyping, fragment sizes for wild-type and mutant alleles are also provided. Table S4 lists new worm strains constructed in the course of this work, as well as previously referenced strains.

DOI: https://doi.org/10.7554/eLife.30789.025

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
