## [Decision Letter]

Thank you for submitting your article "A compartmentalized, self-extinguishing signaling network mediates crossover control in meiosis" for consideration by *eLife*. Your article has been reviewed by three peer reviewers, and the evaluation has been overseen by Bernard de Massy as Reviewing Editor and Jessica Tyler as the Senior Editor. The reviewers have opted to remain anonymous.

The reviewers have discussed the reviews with one another and the Reviewing Editor has drafted this decision to help you prepare a revised submission.

Summary:

The authors have identified three novel members of the ZHP-3 family (named ZHP-1, -2 and -4) in *C. elegans*, in addition to the previously characterized ZHP-3. The authors analyzed the localization of these proteins, their interactions, the consequence of their depletion (using an AID driven degradation strategy) and their functional relationships with other functions involved in CO formation.

Altogether, this manuscript reports a novel and important analysis with the evidence for two distinct activities, one mediated by ZHP-3/4 essential for CO designation and one by ZHP-1/2 limiting ZHP-3/4 activity and promoting chromosome remodeling. Interestingly, ZHP-1/2 are the earliest known factors to localize asymmetrically on chromosome arms after COSA-1 foci formation. The DSB-2 depletion and irradiation experiments are notable advances and lend further support to the idea that recombination sites compete for limiting CO components and/or ZHPs stabilize such a factor. DSB-2 depletion experiments also nicely clarify the relationships between excess recombination sites, ZHP-1/-2, CO designation and ZHP-3/-4 dynamics. This is a very interesting paper that provides roles for three new regulators of meiotic recombination and ties in nicely with recent analyses of RING protein in mice (RNF212 and HEI10), Sordaria and Arabidopsis indicating conserved regulatory pathways.

The data is convincing, and experiments carefully performed. However, the presentation could be improved. Overall the manuscript needs substantial revision.

Several cytological analyses require better presentation and/or additional quantification. Importantly, the authors should revise several of their interpretations and conclusions which are overstated. This also applies to the title which should relate to the experimental evidence not to hypotheses about a proposed self-extinguishing network. It is essential that they integrate and refer to in their interpretations observations from other related studies (see details below). It would also be helpful to clarify the presentation of some analysis due to the complexity of the different effects or markers that are analyzed. It is not clear why the authors presented effects on bivalents and chiasmata first and why they did not start by reporting synapsis, recombination foci…and then chiasmata.

Essential revisions:

Introduction. With respect to mobility of SC proteins, cite pertinent studies by MacQueen (https://www.ncbi.nlm.nih.gov/pubmed/23071451), Villeneuve (https://www.ncbi.nlm.nih.gov/pubmed/28339470) and Colaiacovo (https://www.ncbi.nlm.nih.gov/pubmed/28346135) labs.

Results section. Provide citations for RNF212 and HEI10.

Subsection “Conservation of crossover control mechanisms”. Please note that similar positive-feedback loop ideas have been proposed by others (Villeneuve and Hunter labs) and should be cited.

Subsection “Conservation of crossover control mechanisms”. Cite haplo-insufficiency phenotypes of Rnf212 and Hei10 mice as evidence for roles in crossover assurance.

1) Basic controls: Validation of the signal detected with the antibodies (negative control?).

Validation of the function of the fusion proteins. Are the worms used for cytology homozygous? I assume, Table S1 shows that the transgene are functional. This should be explicitly mentioned.

2) Subsection “ZHP proteins exhibit two distinct patterns of dynamic localization during meiosis”. As this is the first description of ZHP-4, those foci should be quantified as well as their colocalization with COSA-1.

3) Results section. The ZHP signals in the syp-1 mutant should be clarified (late pachytene bright foci).

The authors conclude that ZHP proteins are not associated with chromosomes in syp-1 mutant. This should be demonstrated, which implies that the localization of the polycomplexes with respect to chromosomes should be determined. Several nuclei seem to have 4 foci. Do they colocalize with DSB proteins?

In the syp-1/+ strain, doublets of ZHP-3 are often seen at late pachytene (Figure 1—figure supplement 2C). Is this a reproducible observation?

4) Subsection “Association of ZHPs with meiotic chromosomes depends on synaptonemal complexes”. The authors should define what they mean by CO designation, how it is monitored and what is the basis for assuming that COSA-1 and MSH-5 are required for CO designation.

5) Subsection “ZHPs act as two heterodimeric complexes”. Head of section should be revised. The data presented does not show that the complexes are heterodimeric.

6) Subsection “ZHP-3/4 are essential for CO designation, while ZHP-1/2 limit ZHP-3/4 activity and promote CO maturation”. The dim COSA-1 foci observed in zhp-1/2 mutants should be clarified. This is an essential point since the authors conclude based on this observation that ZHPs limit CO. However, the Spo11 dependency does not tell whether these foci are DSB repair sites. Co-staining with DSB repair protein should be tested (RAD-51, RPA). These foci are obviously not converted into CO. MSH5 foci (Figure 4—figure supplement 1) appear substantially brighter in the zhp-1/2 mutants. Is this correct or a matter of image adjustment/exposure? Do they vary in size/brightness? The increase from mid to late pachytene is not visible on Figure 4—figure supplement 1. Their numbers also appear to be higher than previous various estimates of DSB events.

The authors may consider the possibility that MSH-5 foci are transient or destabilized in zhp-3 (4) mutant.

7) The impact of ZHP-1/2 on axis remodeling is not convincing. The statement that HTP-1 and SYP-1 remain colocalized in zhp-1 mutant (Figure 6—figure supplement 1B) is not seen on the figure (Figure 6—figure supplement 1B) where only a fraction of the proteins colocalizes in nucleus #2.

The authors conclude that in the absence of ZHP-1/2, SC remodeling on axis is defective (with persistence on both short on long arms), but this is not the case in the context of DSB-2. Thus, there is a ZHP-1/2 independent remodeling of SC proteins? Please clarify and revise the statement "ZHP-1/2 mediate chromosome remodeling".

8) Subsection “Conservation of crossover control mechanisms”. Clarify what is meant by "Evidence presented here[…]": Which data indicate that ZHP activity is responsible for signal propagation in crossover interference? How does this idea explain the dosage sensitivity of RNF212/HEI10?

9) Figure 1—figure supplement 2: Western blots: why normalizing by HTP-3 if the level of this protein is not the same in wt and syp1? Tubulin should be used. Then revise subsection “Association of ZHPs with meiotic chromosomes depends on synaptonemal complexes” about the abundance of these proteins.

Provide statistical analysis of the differences in protein levels. Figure 1—figure supplement 2 legend: replace overestimates by underestimates or delete

10) Subsection “ZHPs play essential roles in chiasma formation and chromosome segregation”. It is argued you that zhp-1/2 are more viable than zhp-3, is it statistically significant?

11) Figure 3—figure supplement 1G, H, I, J. For RAD-51 foci, present scatter graphs showing the 4 data points.

12) Figure 3 statistical tests and P values should be provided for the distributions in A-D.

13) Figure 5 panel D, Figure 5—figure supplement 1 Panel C. COSA-1 foci should be quantified

14) Subsection “Conservation of crossover control mechanisms”. SUMO or ubiquitin ligase activities of these proteins have not been established. This statement should be removed.

[Editors' note: further revisions were requested prior to acceptance, as described below.]

Thank you for submitting your article "A compartmentalized signaling network mediates crossover control in meiosis" for consideration by *eLife*. Your article has been reviewed and the evaluation has been overseen by a Reviewing Editor and Jessica Tyler as the Senior Editor.

The reviewers have discussed the reviews with one another and the Reviewing Editor has drafted this decision to help you prepare a revised submission.

We thank the authors for the revised version of the manuscript and their answers to referees’ comments. This revised version provides now a convincing set of data and conclusions.

One important point that needs to be revised in accordance with *eLife* editorial policy is the Abstract. Although the Abstract can certainly highlight the general conclusion about the regulatory network, it must report the key experimental findings (ie the CO controls, designation and maturation, by Zhp-1/Zhp-2 and by Zhp-3 and Zhp-4, the compartmentalization and its asymmetry…).

In addition, several sentences of the abstract should be revised:

It should be acknowledged that one member, Zhp-3, was already identified.

The term "stepwise segregation" should be changed as it will be obscure for most readers.

The use of "winners-takes-all" is not specifically explanatory because one is left with the question of who wins and what does it take to win?

The evolutionary conservation is also not properly reported in this abstract since it was already supported by previous work in mammals, yeast and sordaria.

1) In the main text, change the term « meiotic cohesion » which cannot be understood as such.

2) Subsection “A family of meiotic RING finger proteins in *C. elegans”*, third paragraph. Three data not shown are referred to. Please show homology to mammalian RNF212 and to Hei10 and add the information in Figure 1—figure supplement 1.

3) It would help if the authors emphasize that the compartment they refer to is a bivalent with its SC and thus conveys bivalent autonomous properties. This is mentioned in subsection “ZHP-1/2 mediate chromosome remodeling in response to CO designation”, but it could also be mentioned subsection “Compartmentalization of CO regulation by the synaptonemal complex”.

4) In the Introduction, add "likely" heterodimeric to qualify the protein complexes.

5) Introduction. Whether it is poorly understood is subjective, the authors should rather acknowledge what is known (work on RNF212 and HEI10) or not.

6) Data of the quantification of Zhp-4 and Cosa foci should be provided to support the sentence” At late pachynema, all GFP-COSA-1 foci colocalize with ZHP-4 puncta in 100% of nuclei (n=450 nuclei, 11 rows of nuclei from 8 gonads were scored). 3F indicates 3xFLAG. Scale bars, 5μm.” (Figure 1 legend).

7) Ambiguity persists on the use of CO designation and maturation and the criteria used for monitoring this.

Subsection “ZHP proteins exhibit two distinct patterns of dynamic localization during meiosis”: CO designation is thought to occur at mid pachytene

However, the criteria for CO designation is bright COSA-1 foci which are fully detected at late pachynema (It should be mentioned that the criteria for CO maturation is chiasma or genetic exchange). (Subsection “ZHP proteins exhibit two distinct patterns of dynamic localization during meiosis”: Upon the appearance of GFP COSA-1 foci, which mark designated crossover sites).

The use of « robust » CO designation is also confusing (subsection “ZHP-3/4 are essential to stabilize CO intermediates, while ZHP-1/2 restrict pro-CO activities”) because it is also used to relate to the brightness of COSA-1 foci (Subsection “ZHP-1/2 mediate chromosome remodeling in response to CO designation”: COs were robustly marked by COSA-1).

In Zhp-1 mutant bright COSA-1 foci are absent but still some low level of CO is detected. I assume the authors want to suggest that CO designation is not completely abolished in Zhp-1 mutant (robust is also used to qualify interference and maturation).

---

## [Author Response]

Summary:The authors have identified three novel members of the ZHP-3 family (named ZHP-1, -2 and -4) in C. elegans, in addition to the previously characterized ZHP-3. The authors analyzed the localization of these proteins, their interactions, the consequence of their depletion (using an AID driven degradation strategy) and their functional relationships with other functions involved in CO formation.Altogether, this manuscript reports a novel and important analysis with the evidence for two distinct activities, one mediated by ZHP-3/4 essential for CO designation and one by ZHP-1/2 limiting ZHP-3/4 activity and promoting chromosome remodeling. Interestingly, ZHP-1/2 are the earliest known factors to localize asymmetrically on chromosome arms after COSA-1 foci formation. The DSB-2 depletion and irradiation experiments are notable advances and lend further support to the idea that recombination sites compete for limiting CO components and/or ZHPs stabilize such a factor. DSB-2 depletion experiments also nicely clarify the relationships between excess recombination sites, ZHP-1/-2, CO designation and ZHP-3/-4 dynamics. This is a very interesting paper that provides roles for three new regulators of meiotic recombination and ties in nicely with recent analyses of RING protein in mice (RNF212 and HEI10), Sordaria and Arabidopsis indicating conserved regulatory pathways.The data is convincing, and experiments carefully performed. However, the presentation could be improved. Overall the manuscript needs substantial revision.

We thank the reviewers for their positive comments, and for the very detailed, constructive, and perceptive review. We address the individual comments below.

Several cytological analyses require better presentation and/or additional quantification. Importantly, the authors should revise several of their interpretations and conclusions which are overstated. This also applies to the title which should relate to the experimental evidence not to hypotheses about a proposed self-extinguishing network.

Upon reflection, we agree with the reviewers that other interpretations of our data are equally viable, and we have extensively revised our Discussion section in response and have changed the title of the paper to focus on the compartmentalized nature of the signaling network. We also added additional quantification for several of our cytological analyses and revised some interpretations according the reviewers’ suggestions and comments (see below for details).

It is essential that they integrate and refer to in their interpretations observations from other related studies (see details below). It would also be helpful to clarify the presentation of some analysis due to the complexity of the different effects or markers that are analyzed. It is not clear why the authors presented effects on bivalents and chiasmata first and why they did not start by reporting synapsis, recombination foci…and then chiasmata.

We have modified the order of presentation, and feel that the current flow, in which the localization of the proteins is described, and their functions were then analyzed systematically, was the most straightforward presentation. In particular, we feel that the observation that chiasmata depend absolutely on ZHP-3/4 but only partially on ZHP-1/2 helps to contextualize and interpret some of the findings we describe subsequently regarding recombination intermediates.

Essential revisions:Introduction. With respect to mobility of SC proteins, cite pertinent studies by MacQueen (https://www.ncbi.nlm.nih.gov/pubmed/23071451), Villeneuve (https://www.ncbi.nlm.nih.gov/pubmed/28339470) and Colaiacovo (https://www.ncbi.nlm.nih.gov/pubmed/28346135) labs.Results section. Provide citations for RNF212 and HEI10.Subsection “Conservation of crossover control mechanisms”. Please note that similar positive-feedback loop ideas have been proposed by others (Villeneuve and Hunter labs) and should be cited.Subsection “Conservation of crossover control mechanisms”. Cite haplo-insufficiency phenotypes of Rnf212 and Hei10 mice as evidence for roles in crossover assurance.

We thank the reviewers for this suggestion. We have added the suggested references (some of which were not published at the time of our initial submission) and related work from other groups in the new manuscript.

1) Basic controls: Validation of the signal detected with the antibodies (negative control?).Validation of the function of the fusion proteins. Are the worms used for cytology homozygous? I assume, Table S1 shows that the transgene are functional. This should be explicitly mentioned.

We thank the reviewers for this suggestion. The antibodies used in this study have all been previously validated for specificity in *C. elegans* (appropriate references are cited in the Materials and methods section), with the possible exception of the well-characterized commercial anti-epitope antibodies used to detect tagged proteins. To address the reviewers’ comments, we now include images demonstrating a lack of background staining with these anti-epitope antibodies in *C. elegans* germline tissue (Figure 1—figure supplement 2 and Figure 2—figure supplement 2A-B). We also state explicitly in the manuscript that all transgenes used in the study are fully functional, as documented quantitatively in Supplementary file 1. Except where otherwise stated, animals used for cytological analysis were homozygous for mutant or degron-tagged alleles. In all cases, the genotypes are specified in the text, figures, and/or figure legends.

2) Subsection “ZHP proteins exhibit two distinct patterns of dynamic localization during meiosis”. As this is the first description of ZHP-4, those foci should be quantified as well as their colocalization with COSA-1.

Quantification of immunofluorescent foci, particularly in whole-mount tissues, is inherently challenging and problematic. Differences in the affinity of antibodies or abundance of various proteins make it problematic to compare numbers of foci, since an unknown fraction of foci may be below the limit of detection. For example, we report here that we detect more MSH-5 foci than COSA-1 foci per nucleus in both wild-type and *zhp-1/2* mutants, but this likely reflects a difference in sensitivity rather than a *bona fide* difference in the frequency of their association with recombination intermediates. We have quantified foci in those cases where we felt it would be meaningful to compare between genotypes.

In response to the reviewers’ specific request regarding ZHP-4 foci, we have now quantified the colocalization of ZHP-4 with COSA-1. We report that in 100% of wild-type oocytes, all GFP-COSA-1 foci at late pachytene colocalized with ZHP-4 puncta (11 row of nuclei from every 8 gonads were scored). We include this data in the revised manuscript. However, we feel quantification of ZHP-4 foci in early prophase would be a very difficult task and would not be informative.

3) Results section. The ZHP signals in the syp-1 mutant should be clarified (late pachytene bright foci).The authors conclude that ZHP proteins are not associated with chromosomes in syp-1 mutant. This should be demonstrated, which implies that the localization of the polycomplexes with respect to chromosomes should be determined. Several nuclei seem to have 4 foci. Do they colocalize with DSB proteins?In the syp-1/+ strain, doublets of ZHP-3 are often seen at late pachytene (Figure 1—figure supplement 2C). Is this a reproducible observation?

We thank the reviewers for the question. We have now added statements to clarify that these foci of ZHP proteins do not colocalize with SYP proteins, and are thus not polycomplexes, but rather a distinct type of aggregate. We have also added images to show that ZHPs foci in *syp-1* mutants do not colocalize with recombination intermediates marked by RAD-51 (Figure 1—figure supplement 3D).

We have examined our 3D data stacks in detail and conclude that doublets of ZHP-3 staining are infrequently observed. Given that we did not observe nuclei at later stages showing more than 6 ZHP-3 foci, we do not think it likely that occasional closely spaced ZHP-3 foci represent multiple sites along the same chromosome.

4) Subsection “Association of ZHPs with meiotic chromosomes depends on synaptonemal complexes”. The authors should define what they mean by CO designation, how it is monitored and what is the basis for assuming that COSA-1 and MSH-5 are required for CO designation.

We thank the reviewers for pointing out these ambiguities in our terminology. Prior work has established that COSA-1, ZHP-3, and MutSƔ are not required for synapsis in *C. elegans* but are all required for the appearance of bright COSA-1 foci along chromosomes at mid-pachytene, which have been convincingly established to correspond to sites of designated crossovers. Operationally, crossover designation is thus defined as the appearance of bright COSA-1 foci after mid-pachytene. COSA-1 and MutSƔ are also interdependent for the appearance of early foci that stain faintly with MSH-5 antibodies (MSH-4 antibodies and tagged transgenes have not produced signals above the limit of our detection). In the revised manuscript we show that ZHP-3/4 are also required for detection of these early, dim foci. Given that the biochemical basis for this designation remains uncertain, we have now largely eliminated the phrase “crossover designation,” and instead used more descriptive phrases such as “bright COSA-1 foci,” throughout the manuscript. Evidence from other organisms suggests that crossover designation and crossover resolution/maturation may be temporally distinct processes, so we have also avoided the use of the term “crossover formation” to describe the sites marked by COSA-1 foci.

5) Subsection “ZHPs act as two heterodimeric complexes”. Head of section should be revised. The data presented does not show that the complexes are heterodimeric.

We have provided complementary lines of evidence that these proteins act as heterodimers: in addition to showing that the four ZHPs show pairwise interdependence for their localization and function, and in some cases for stability, we provide physical interaction data based on yeast 2-hybrid analysis, which is generally regarded as reporting on direct protein-protein interactions (Fields and Song, (1989). A novel genetic system to detect protein-protein interactions PMID: 2547163 DOI:10.1038/340245a0). Based on ample precedents for dimerization of RING finger E3 ligases, our evidence supports the idea that these proteins function as heterodimers or heterooligomers.

However, to soften the statement, we have revised the section heading to read: “ZHPs likely act as two heterodimeric complexes.”

6) Subsection “ZHP-3/4 are essential for CO designation, while ZHP-1/2 limit ZHP-3/4 activity and promote CO maturation”. The dim COSA-1 foci observed in zhp-1/2 mutants should be clarified. This is an essential point since the authors conclude based on this observation that ZHPs limit CO. However, the Spo11 dependency does not tell whether these foci are DSB repair sites. Co-staining with DSB repair protein should be tested (RAD-51, RPA). These foci are obviously not converted into CO. MSH5 foci (Figure 4—figure supplement 1) appear substantially brighter in the zhp-1/2 mutants. Is this correct or a matter of image adjustment/exposure? Do they vary in size/brightness? The increase from mid to late pachytene is not visible on Figure 4—figure supplement 1. Their numbers also appear to be higher than previous various estimates of DSB events.The authors may consider the possibility that MSH-5 foci are transient or destabilized in zhp-3 (4) mutant.

We thank the reviewers for highlighting this issue, and we provide additional information to clarify this in the revised manuscript. Specifically, we have now co-stained for COSA-1 and RAD-51 (Author response image 1) and for COSA-1 and MSH-5 (Figure 4—figure supplement 1E) in *zhp-1* mutants. We find that COSA-1 foci do not colocalize with RAD-51 (as is also true for wild-type *C. elegans*). However, we found that dim COSA-1 foci do colocalize extensively with MSH-5. Together with our findings that these foci are absent or greatly reduced in *spo-11* and *dsb-2* mutants, respectively, this supports the idea that dim COSA-1 foci correspond to recombination intermediates.

Indeed, MSH-5 foci (Figure 4—figure supplement 1) appear substantially brighter in the *zhp-1/2* mutants, which likely reflects a failure of the “winner takes all” nature of the ZHP circuit. The increase in MSH-5 intensity from mid to late pachytene is much more obvious in *zhp-1* mutants than in WT. The brightness of MSH-5 foci can vary markedly in the same nucleus. While we have quantified the relative numbers of these foci in *zhp-1/2* vs. wild-type animals, we do not feel that these can be regarded as absolute numbers of intermediates, due to the limits of fluorescent detection, particularly for proteins of low abundance (as argued above). We also now report that early MSH-5 and COSA-1 foci are not detected in *zhp-3/4* mutants and suggest that this may be due to a failure of stabilization.

Quantification of DSBs in *C. elegans* has been challenging and contentious. Again, different markers (e.g., MSH-5 vs. RAD-51) have different sensitivities of detection. Additionally, mutations that block the repair of DSBs in *C. elegans* (e.g., *rad-54*) lead to a prolonged period of DSB formation and an increase in their number, making it very difficult to assess this question, and we are not aware of any reliable estimates for the number of DSBs in wild-type animals. (See our discussion of this issue in Yu et al., 2016). We are working to address this important question by immunoprecipitating SPO-11-bound oligonucleotides.

7) The impact of ZHP-1/2 on axis remodeling is not convincing. The statement that HTP-1 and SYP-1 remain colocalized in zhp-1 mutant (Figure 6—figure supplement 1B) is not seen on the figure (Figure 6—figure supplement 1B) where only a fraction of the proteins colocalizes in nucleus #2.The authors conclude that in the absence of ZHP-1/2, SC remodeling on axis is defective (with persistence on both short on long arms), but this is not the case in the context of DSB-2. Thus, there is a ZHP-1/2 independent remodeling of SC proteins? Please clarify and revise the statement "ZHP-1/2 mediate chromosome remodeling".

We recognize that it can be difficult, particularly for researchers unfamiliar with meiotic cytology in *C. elegans*, to appreciate and interpret the differences in bivalent structure we document for various genotypes. We have tried to clarify the text and figure legends to make the images easier to interpret. Our key findings are that HTP-1 and SYP-1 fail to show reciprocal localization on the few bivalent chromosomes present in *zhp-1/2* mutants, and also colocalize along univalent chromosomes. In contrast, in other mutants lacking COs, univalent chromosomes are usually positive for HTP-1 or SYP-1 staining, but not both (Martinez-Perez et al., ‎2005).

We used *zhp-1/2; dsb-2* double mutants to analyze this issue further, since in these animals we do observe occasional bright COSA-1 foci at late pachytene. These foci correspond to the number of bivalents observed at diakinesis, indicating that they mature as crossovers. Although ZHP-3/4 were depleted from the corresponding SCs, other hallmarks of remodeling (e.g., PLK-2 localization and SC retention along the short arms) were defective, leading to the hierarchical view of remodeling described in the text and in Figure 8.

We hope that these clarifications fully address the reviewers’ concerns.

8) Subsection “Conservation of crossover control mechanisms”. Clarify what is meant by "Evidence presented here[…]": Which data indicate that ZHP activity is responsible for signal propagation in crossover interference? How does this idea explain the dosage sensitivity of RNF212/HEI10?

We have removed this statement, since we agreed that it was confusing and that other interpretations are equally consistent with our data. We address the dosage sensitivity of RNF212/HEI10 in the Discussion section.

9) Figure 1—figure supplement 2: Western blots: why normalizing by HTP-3 if the level of this protein is not the same in wt and syp1? Tubulin should be used. Then revise subsection “Association of ZHPs with meiotic chromosomes depends on synaptonemal complexes” about the abundance of these proteins.Provide statistical analysis of the differences in protein levels. Figure 1—figure supplement 2 legend: replace overestimates by underestimates or delete

We thank the reviewers for pointing out this. We routinely load lysate from an equal number of age-matched worms in each lane, which is probably at least as useful as other normalization methods. The abundance of many proteins, including tubulin, is affected by meiotic mutations such as *syp-1*, presumably due to differential abundance of proliferating and meiotic stages in the germline. Using tubulin and HTP-3 to normalize our ZHP blots gave very similar results (Author response image 2). We chose to include HTP-3 as reference, since this protein is restricted to nuclei in meiotic prophase, while tubulin is present in all germline and somatic cells. Because both HTP-3 and tubulin are more abundant in *syp-1* mutants, our normalized measurements likely underestimate the degree to which the ZHP proteins are reduced/destabilized in the absence of synapsis. However, this is not a central finding of this work.

**Author response image 2. respfig2:** 

10) Subsection “ZHPs play essential roles in chiasma formation and chromosome segregation”. It is argued you that zhp-1/2 are more viable than zhp-3, is it statistically significant?

Yes, it is clearly significant. We have added p values based on Student’s *t*-test in the revised manuscript.

11) Figure 3—figure supplement 1G, H, I, J. For RAD-51 foci, present scatter graphs showing the 4 data points.

We have replaced the previous bar graphs by the new scatter graphs (Figure 3—figure supplement 3C-F in the new version of manuscript). However, we think that the bar graphs are actually easier to read, so we would like to leave the bar graphs in this response letter (Author response image 3).

**Author response image 3. respfig3:** 

12) Figure 3 statistical tests and P values should be provided for the distributions in A-D.

We have now analyzed these distributions using the Chi-square test. The results and p values are provided in the figure legend.

13) Figure 5 panel D, Figure 5—figure supplement 1 Panel C. COSA-1 foci should be quantified

We thank the reviewers for this suggestion. However, we feel that this would be impossible to do in a meaningful way due to the sensitivity issues raised above. As we describe, as we exposed animals to increasing doses of radiation, the number of dim foci increased, and bright foci decreased. However, the brightness of the foci in the same nuclei varies widely, making it impossible to classify them. Additionally, it should be noted that all published COSA-1 quantification to date has used a bombarded (multicopy) GFP-COSA-1 transgene, which is likely overexpressed relative to wild-type COSA-1 and may therefore not represent the wild-type situation. That said, we have quantified COSA-1 foci in Figure 5 panel A, which strongly support our conclusion. Figure 5 panel D was included to document the gradual change in both brightness and number of COSA-1 foci, since the images provide a better understanding than any quantification.

14) Subsection “Conservation of crossover control mechanisms”. SUMO or ubiquitin ligase activities of these proteins have not been established. This statement should be removed.

We agree that there are no published findings that convincingly establish SUMO or ubiquitin ligase activity for the Zip3/RNF212/Hei10/ZHP family. However, interpretations in several prominent publications have led many in the field to accept the idea that these proteins have opposing SUMO and ubiquitin ligase activity. We thus feel that it is relevant and important to highlight prior findings from *C. elegans* that indicate that SUMOylation is dispensable for crossover control in this organism. We have removed comments about the likely activities of homologs in other organisms.

[Editors' note: further revisions were requested prior to acceptance, as described below.]

One important point that needs to be revised in accordance with eLife editorial policy is the Abstract. Although the Abstract can certainly highlight the general conclusion about the regulatory network, it must report the key experimental findings (ie the CO controls, designation and maturation, by Zhp-1/Zhp-2 and by Zhp-3 and Zhp-4, the compartmentalization and its asymmetry…).In addition, several sentences of the abstract should be revised:It should be acknowledged that one member, Zhp-3, was already identified.The term "stepwise segregation" should be changed as it will be obscure for most readers.The use of "winners-takes-all" is not specifically explanatory because one is left with the question of who wins and what does it take to win?The evolutionary conservation is also not properly reported in this abstract since it was already supported by previous work in mammals, yeast and sordaria.

We thank the reviewer for these suggestions. We have now revised the Abstract extensively to address these comments. Specifically, we now mention the conserved role of the ZHP family of proteins, the prior evidence that ZHP-3 is required for crossovers, and compartmentalization of these factors within the synaptonemal complex. To include these ideas while keeping the Abstract within 150 words, it was necessary to remove some of the other statements. We have deleted the sentence regarding the winner-takes-all nature of the circuit, since it requires more space than available in the Abstract to clarify, but we do still mention this idea in the Discussion section. (We note that “winner-take-all” is a well-established term that describes a class of regulatory mechanisms). We also removed the phrase “stepwise segregation” from the Abstract, for similar reasons.

1) In the main text, change the term « meiotic cohesion » which cannot be understood as such.

The reviewers have not explained why they object to this term, but to address this comment we have now replaced the phrase “meiotic cohesion” with “sister chromatid cohesion in meiosis” or similar alternatives throughout the manuscript.

2) Subsection “A family of meiotic RING finger proteins in C. elegans”, third paragraph. Three data not shown are referred to. Please show homology to mammalian RNF212 and to Hei10 and add the information in Figure 1—figure supplement 1.

We appreciate this suggestion. However, we made a decision not to include alignments, phylogenetic trees, or other visual evidence for homology between ZHPs, Zip3, RNF212, and HEI10 because we find that these alignments are highly sensitive to the specific algorithms and parameters chosen. Unsurprisingly, the well-conserved RING finger domains dominate any alignments among these proteins. Coiled-coil domains are notoriously refractory to homology recognition and alignment because they have a highly skewed amino acid composition. The highly divergent, probably unstructured, C-terminal domains likely confer the unique activities of these proteins, but are difficult to align meaningfully, in part because such domains do not conform to the same patterns of amino acid substitutions as globular protein domains. The sequences of the ZHP proteins and their homologs are readily available, so interested readers can easily and quickly verify homology by doing their own BLAST or other homology searches and alignments, using the tools they prefer.

3) It would help if the authors emphasize that the compartment they refer to is a bivalent with its SC and thus conveys bivalent autonomous properties. This is mentioned in subsection “ZHP-1/2 mediate chromosome remodeling in response to CO designation”, but it could also be mentioned subsection “Compartmentalization of CO regulation by the synaptonemal complex”.

By "compartment” we actually mean the SC that assembles between a pair of chromosomes (or an individual polycomplex), rather than the bivalent as a whole. We have added additional comments in several places, including the abstract, to clarify our use of this term.

4) In the Introduction, add "likely" heterodimeric to qualify the protein complexes.

We have added “likely” to qualify this statement. We note that the reviewers have not explained their skepticism about the idea that ZHP-1/2 and ZHP-3/4 are heterodimeric complexes, given that we provide extensive physical and functional interaction data and there are many precedents for dimerization of RING finger proteins.

5) Introduction. Whether it is poorly understood is subjective, the authors should rather acknowledge what is known (work on RNF212 and HEI10) or not.

We have removed the comment.

6) Data of the quantification of Zhp-4 and Cosa foci should be provided to support the sentence “At late pachynema, all GFP-COSA-1 foci colocalize with ZHP-4 puncta in 100% of nuclei (n=450 nuclei, 11 rows of nuclei from 8 gonads were scored). 3F indicates 3xFLAG. Scale bars, 5μm.” (Figure 1 legend).

We do not understand what the reviewers are requesting here. Figure 1 includes representative images showing colocalization of COSA-1 with ZHP-3 and ZHP-4 at late pachytene. The legend describes our quantification of the extent of colocalization between COSA-1 and ZHP-4 at this stage (as previously requested by the reviewers), including the numbers of nuclei and gonads analyzed. We felt that including this statement in the main text disrupted the flow, so we chose to put it in the figure legend, but we can move it to the main text if preferred.

7) Ambiguity persists on the use of CO designation and maturation and the criteria used for monitoring this.

We have now replaced most of the remaining instances of the phrase “CO designation” with the more objective “accumulation of pro-CO factors at a single site” or similar.

Subsection “ZHP proteins exhibit two distinct patterns of dynamic localization during meiosis”: CO designation is thought to occur at mid pachyteneHowever, the criteria for CO designation is bright COSA-1 foci which are fully detected at late pachynema (It should be mentioned that the criteria for CO maturation is chiasma or genetic exchange). (Subsection “Subsection “ZHP proteins exhibit two distinct patterns of dynamic localization during meiosis”: Upon the appearance of GFP COSA-1 foci, which mark designated crossover sites).

In this particular context, our statements summarize the conclusions of Yokoo et al., (2012). While COSA-1 foci normally become brighter from mid to late pachytene, the accumulation of COSA-1 at a single site initiates by mid-pachytene, indicating that the site has been selected/designated at that time.

The use of « robust » CO designation is also confusing (subsection “ZHP-3/4 are essential to stabilize CO intermediates, while ZHP-1/2 restrict pro-CO activities”) because it is also used to relate to the brightness of COSA-1 foci (Subsection “ZHP-1/2 mediate chromosome remodeling in response to CO designation”: COs were robustly marked by COSA-1).In Zhp-1 mutant bright COSA-1 foci are absent but still some low level of CO is detected. I assume the authors want to suggest that CO designation is not completely abolished in Zhp-1 mutant (robust is also used to qualify interference and maturation).

We have removed the adjective “robust” from our descriptions of the crossover designation process, and we have also eliminated the phrase “crossover designation,” as noted above. It remains unclear why some COs are detected (by both genetic recombination analysis and the presence of bivalents at diakinesis) in *zhp-1/2* mutants, even though we do not observe accumulation of pro-CO factors at a subset of recombination intermediates. It may be that sufficient pro-CO factors stochastically accumulate at a subset of sites to promote CO maturation through the Class I pathway, or these COs may instead be Class II COs, which have been observed in some *C. elegans* mutants. This ambiguity makes it difficult to say whether “CO designation” in the presence of excess DSBs strictly depends on *zhp-1/2*. Our experiments with *dsb-2* mutants indicate that the roles of *zhp-1/2* are less essential for accumulation of pro-CO factors at recombination intermediates when competition between these intermediates is reduced or eliminated.